# On the effects of similarity metrics in decentralized deep learning under distributional shift

**Edvin Listo Zec**                                                          *edvin.listo.zec@ri.se*
*RISE Research Institutes of Sweden*
*KTH Royal Institute of Technology*

**Tom Hagander**
*Lund University*

**Eric Ihre-Thomason**
*Lund University*

**Sarunas Girdzijauskas**
*KTH Royal Institute of Technology*

**Reviewed on OpenReview:** *https://openreview.net/forum?id=WppTEs4Kkn*

## Abstract

Decentralized Learning (DL) enables privacy-preserving collaboration among organizations or users to enhance the performance of local deep learning models. However, model aggregation becomes challenging when client data is heterogeneous, and identifying compatible collaborators without direct data exchange remains a pressing issue. In this paper, we investigate the effectiveness of various similarity metrics in DL for identifying peers for model merging, conducting an empirical analysis across multiple datasets with distribution shifts. Our research provides insights into the performance of these metrics, examining their role in facilitating effective collaboration. By exploring the strengths and limitations of these metrics, we contribute to the development of robust DL methods.

## 1 Introduction

In recent years, there has been a significant increase in interest towards distributed machine learning approaches, driven primarily by growing concerns over user privacy and data security. This shift is driven by regulations such as the California Consumer Privacy Act (CCPA), General Data Protection Regulation (GDPR), and the Data Protection Act, alongside a rising awareness among users about their data rights. These developments necessitate advancements in privacy-preserving methods. Additionally, many organizations possess proprietary data that cannot be shared externally, further fueling the need for decentralized frameworks.

As machine learning models become more sophisticated and continue to scale, they exhibit superior performance across a diverse array of tasks. Yet, these models rely heavily on extensive datasets, often compiled from various online sources. This poses significant issues regarding data ownership and privacy. Therefore, decentralized approaches are critical in mitigating these issues by enabling collaborative model training without the necessity for a centralized dataset repository. In this framework, no single entity controls a vast dataset. Instead, multiple stakeholders participate in model training through a decentralized and confidential approach.

Among these approaches, federated learning (FL) has emerged as a leading framework for distributed deep learning, allowing the training of a global model across multiple clients that maintain private datasets (McMahan et al., 2017). Various studies have aimed to address *client drift*—a phenomenon occurring when

clients pursue divergent optimization objectives due to distributional shifts between datasets, resulting in diminished performance upon averaging their model parameters. To counter this, several strategies introduce a regularization objective into the global training goal to minimize deviations between the global model and individual local models (Li et al., 2020; Karimireddy et al., 2020; Li et al., 2021). This approach is effective when clients share similar datasets, sampled from the same underlying data generating process. In these cases, a common global model is optimal. Conversely, in scenarios where clients' data originate from distinct generating processes, these methods can be suboptimal and, in some cases, detrimental. Ghosh et al. (2020); Vardhan et al. (2024) address this issue within centralized federated learning by training multiple global models—specifically, $K$ models corresponding to the number of data clusters, with clustering determined by empirical loss to approximate data similarity. Similarly, Sattler et al. (2020) employ cosine similarity on local gradients for clustering clients in FL. Although effective, this leads to a high computational cost as the similarity is computed at stationary solutions of the FL objective. Moreover, there is other notable work on clustered centralized FL which also consider cosine similarity or Euclidean distance between parameters (Briggs et al., 2020; Duan et al., 2021; Ruan & Joe-Wong, 2022).

## 1.1 Technical challenges and related work

The aforementioned approaches necessitate a central server. In contrast, our work explores fully decentralized methods for identifying client clusters, which are more suitable for large-scale applications. While federated learning enables distributed training of a global model, its reliance on a central server can become a limiting factor and a bottleneck as the number of clients grows. This limitation has spurred interest in fully decentralized systems, known as decentralized learning (DL), where clients interact within a peer-to-peer network, eliminating the need for a central server. Decentralized learning is particularly advantageous for its scalability and inherent robustness against single-point failures or targeted attacks. An illustration of different machine learning frameworks is shown in Figure 1.

One notable approach within decentralized learning is gossip learning, a peer-to-peer communication protocol, which has been extensively studied in various machine learning contexts (Kempe et al., 2003; Boyd et al., 2006; Ormándi et al., 2013; Hegedűs et al., 2019). Initial decentralized work using gossip-based optimization for non-convex deep learning tasks, specifically with convolutional neural networks (CNNs), was explored by Blot et al. (2016). However, gossip learning faces challenges in non independent and identically distributed (non-iid) settings where there is a distributional shift across client datasets. This is primarily due to the inadequacy of gossip learning in selectively identifying beneficial peers during the training process. To address these challenges, recent work has addressed decentralized learning within non-iid contexts. Onoszko et al. (2021); Li et al. (2022) introduced client selection algorithms based on empirical loss, employing distinct discovery and selections phases. These studies address challenges in decentralized learning within non-iid settings but require extensive hyperparameter tuning, particularly regarding search length, and rely on hard cluster assignments for clients. Such hard clustering can be detrimental if the discovery phase is noisy. To mitigate these issues, Listo Zec et al. (2022) proposed a sampling strategy that converts the similarity scores into sampling probabilities, which are used each communication round to sample collaborators. This approach identifies similar clients and facilitates model merging in proportion to their similarity. By adopting a soft clustering technique, it enhances the efficiency of peer collaboration, circumventing the need for the computationally intensive and difficult-to-tune discovery phase required by hard clustering methods. Additionally, Listo Zec et al. (2024) introduced a framework using multi-armed bandits for client selection, where the selection process is modeled on empirical rewards derived from validation accuracy.

Quasi-Global Momentum (QGM) (Lin et al., 2021), Neighbourhood Gradient Mean (NGM) (Aketi et al., 2023) and Data-Heterogeneity-Aware Mixing (DHM) (Dandi et al., 2022) are contemporary approaches that address the challenges posed by non-iid data in decentralized learning. While the methods focus on improving consensus optimization, our work diverges significantly in its objectives and assumptions. Specifically, we aim to tackle the problem of varying data distributions across clusters rather than optimizing for a shared consensus model.

## 1.2 Our contributions

Previous research has predominantly employed empirical loss as a proxy for data similarity (Ghosh et al., 2019; Onoszko et al., 2021; Listo Zec et al., 2022), while some studies have investigated similarity using gradients (Li et al., 2022; Sattler et al., 2020). Despite significant progress, the impact of the chosen similarity metric on decentralized learning, especially under distribution shifts between client datasets, remains insufficiently explored. In this paper, we fill this gap by thoroughly study different similarity metrics in decentralized learning through extensive empirical experiments on common benchmark datasets.

In summary, a critical question persists which we seek to answer: *how does the choice of similarity metric influence client identification and the performance of decentralized learning?*

This question is crucial because the effectiveness of decentralized learning heavily relies on accurately identifying and engaging similar peers. To address this gap, we present a comprehensive empirical study exploring the impacts of four distinct similarity metrics: empirical loss, cosine similarity on gradients, cosine similarity on model weights, and $L^2$ distance on model weights.

Furthermore, we introduce a novel aggregation method for decentralized learning called Federated Similarity Averaging (FedSim). Unlike the traditional Federated Averaging (FedAvg) approach, which aggregates models based on training set sizes, FedSim merges client models using a weighted average based on their similarities. This approach allows clients to reduce the influence of dissimilar peers on their local model, thereby mitigating the impact of incorrectly sampled clients. In our experiments, we compare FedSim against FedAvg across various similarity metrics and demonstrate that in several scenarios, FedSim can significantly enhance performance.

This study not only fills a gap in understanding the role of similarity metrics in decentralized learning but also offers practical insights into optimizing collaboration in environments subject to distributional shifts. Our experimental results show that empirical loss can be a noisy estimator of data similarity in many cases, especially when the number of samples is small, leading to subpar client identification and negatively affecting performance. At the same time, we show that there are other metrics that can be more robust. By systematically evaluating these metrics, we provide a deeper understanding of their influence on the performance and efficiency of decentralized learning systems, hopefully guiding future research and application in this domain.

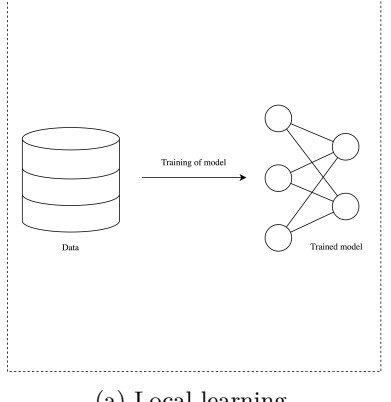
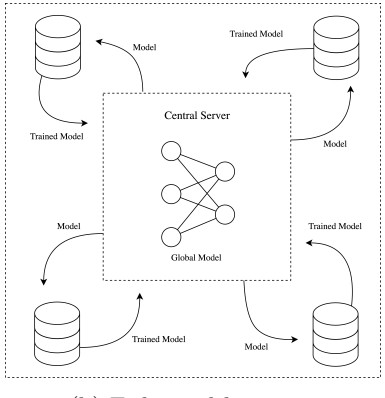
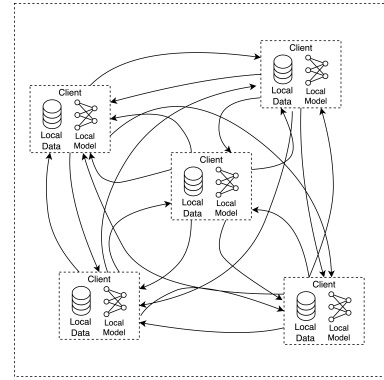

(a) Local learning.    (b) Federated learning.    (c) Decentralized learning.

Figure 1: Illustrations of different types of machine learning paradigms. (a) In traditional ML, a model is trained on a centralized dataset. (b) In FL, a central server orchestrates the training of a global model across multiple private datasets. (c) In DL, there is no central server; instead clients communicate and merge models in a peer-to-peer network.

## 2 Problem formulation

Consider a set of clients denoted by $\mathcal{S}$, each possessing a local and **private** dataset $\mathcal{D}_i = \{(x_n, y_n)\}_{n=1}^{N_i}$ with $N_i$ samples, sampled from an underlying probability distribution $P_i(x, y)$ for $i \in \mathcal{S}$. In decentralized learning, we have a network of $K = |\mathcal{S}|$ clients, each with a local optimization objective consisting of a loss function $\ell$ over model parameters $w_i$. The primary goal for each client $i$ is to learn an optimal model $f_i^\star$ parameterized by $w_i^\star$, which minimizes the following objective:

$$R(w_i) = \int \ell(f_i(w_i; x), y) dP_i(x, y). \tag{1}$$

Given that the true distribution $P(x, y)$ is unobserved, direct optimization of equation 1 is not possible. Instead, each client performs empirical risk minimization (ERM) locally by minimizing

$$\hat{R}(w_i) = \mathbb{E}_{(x,y) \sim \mathcal{D}_i} \left[ \ell(f_i(w_i; x), y) \right], \tag{2}$$

to obtain $w_i^\star = \arg\min_{w_i \in \mathbb{R}^d} \hat{R}(w_i)$.

However, the sample size $N_i$ of each local dataset $\mathcal{D}_i$ is assumed to be insufficient for solving equation 2. Consequently, clients need to collaborate by merging models with peers to solve their local optimization problems. For any client $i$, each communication round $t$ starts with $E$ local epochs of training where the local model is updated using an optimizer, such as stochastic gradient descent (SGD). Then, client $i$ interacts with a random subset of clients $\mathcal{M}^t \subset \mathcal{S}$, merging models through federated averaging (McMahan et al., 2017). This process involves calculating a weighted average of the model parameters, resulting in an updated model:

$$w^{t+1} = \sum_{i=1}^{m} \frac{N_i}{N} w_i^t, \tag{3}$$

where $N = \sum_{i=1}^{m} N_i$ and $m = |\mathcal{M}|$. After this aggregation, local training resumes for $E$ epochs, and the process repeats for $T$ rounds. In summary, there are $K$ models in the communication network and each client uses this collaborative approach to enhance model performance locally while preserving data privacy.

In standard federated and decentralized learning, it is implicitly assumed that a single model can effectively fit all client data simultaneously. This assumption holds true in an ideal scenario where all clients share the same data generating process, i.e., $P_i = P_j$ for all $i, j \in \mathcal{S}$. In such cases, decentralized learning with random communication (also known as gossip learning) is effective. However, when clients form clusters with distinct data generating processes, i.e., $P_i \neq P_j$, merging models from clients with different distributions can significantly hinder convergence or even be detrimental (Zhao et al., 2018; Kairouz et al., 2021). This is because the objective function in equation 1 varies between clients, and there may not be a single model $w_i^\star$ that achieves zero risk across all distributions. Thus, the challenge for each client is to identify subsets of clients with similar data distributions to ensure that merging models is beneficial, all while preserving the privacy of the datasets.

### 2.1 Similarity based peer selection

In our problem setup, we assume that the number of unique data distributions $P_k$ is fewer than the total number of clients $K$, implying certain devices share data generation processes and form clusters. Thus, each client's task is to identify peers within their respective clusters with whom they can merge models to optimize their local objectives while preserving data privacy. This task essentially becomes a combinatorial selection problem where each client must select $m$ peers from $\binom{K}{m}$ possible combinations during each communication round. The objective for each client is to select the most appropriate $m$ peers each round to minimize its local objective function. When all client datasets are sampled from the same joint probability distribution $P_k(x, y)$ and are partitioned independently and identically distributed (iid), random communication – i.e., uniform sampling of $\mathcal{M}^t \subset \mathcal{S}$ – is effective. However, the goal of similarity-based sampling is to strategically select the set $\mathcal{M}^t$ and merge models in a manner that enables each client to minimize the objective specified in equation 1. This approach aims to enhance the efficiency and performance of decentralized learning by leveraging the inherent similarities between clients' data distributions.

Since local datasets are private, traditional measures cannot be used to calculate similarities between them directly for clustering clients. Instead, we rely on model parameters or gradients to infer similar datasets. To maintain simplicity, we adopt the decentralized adaptive clustering (DAC) framework (Listo Zec et al., 2022). In this framework, each client $i$ stores a probability vector $\mathbf{p_i^t} \in \mathbb{R}^K$ which is used to sample new clients $\mathcal{M}^t$, where each element $p_{ij}^t$ represents the probability for client $i$ to sample client $j$ at time $t$. This probability vector is calculated using a softmax function and a similarity metric $\mathbf{s_i}$ every time step $t$, as follows:

$$\mathbf{p_i}(\mathbf{s_i}) = \frac{e^{\tau \mathbf{s_i}}}{\sum_{k=1}^{K} e^{\tau s_{ik}}}. \tag{4}$$

Here, $\tau$ is the inverse temperature parameter controlling the distribution of the softmax, i.e. $\tau = 0$ gives a uniform distribution and as $\tau \to \infty$ we reach the argmax function. This is a hyperparameter that is problem dependent and that needs to be optimized.

The selection of similarity metric $\mathbf{s_i}$ is critical in the DAC framework as it influences how effectively similar clients are identified and, consequently, how efficiently the models converge. Listo Zec et al. (2022) used inverse empirical loss as a similarity metric, defined by evaluating client $i$'s model $w_i^t$ at time $t$ on the training data of client $k$, expressed as $s_{ik}^t = \left( \sum_{x_k, y_k \in \mathcal{D}_k} \ell(w_i^t; x_k, y_k) \right)^{-1}$. This metric can be computed without sharing data samples, as only model parameters and similarity scores are exchanged between clients $i$ and $k$. The authors demonstrated that this method effectively identifies similar clients over time in decentralized learning. However, this metric may have drawbacks since its performance depends on sample size and quality. Additionally, the authors did not compare it with other similarity metrics within this framework. Therefore, our work aims to explore how different similarity metrics impact client selection and performance in DL.

Similar to previous studies, this work assumes a fully connected network where all clients can communicate directly with each other, facilitated by communication protocols such as those available via the internet. We also assume synchronous client communication, as our focus is not on system heterogeneity. While exploring the impact of constrained network topologies or varying levels of client connectivity on performance is a compelling area for future research, it is beyond the scope of this paper.

---

**Algorithm 1** Decentralized learning protocol for a client $i$

---

1: **Input**: model parameters $w_i$, temperature $\tau$, learning rate $\eta$, dataset $x_i, y_i \sim \mathcal{D}_i$.
2: Initialize prior probabilities: $p_{ij} = \frac{1}{K-1}$
3: **for** each communication round $t = 1, \dots, T$ **do**
4:     Select $m \leq K - 1$ clients $\mathcal{M}_i^t = \{c_1, c_2, \dots, c_m\}$ without replacement, according to the probability distribution $\mathbf{p}_i^t = [p_{i,1}^t, p_{i,2}^t, \dots, p_{i,K-1}^t]$.
5:     Compute similarity scores $s_{ij}^t$ between client $i$ and each sampled client $j \in \mathcal{M}_i^t$.
6:     Update probability vector via softmax: $\mathbf{p}_i^t \leftarrow \texttt{SOFTMAX}(\mathbf{s}_i^t, \tau)$.
7:     For FedSim, normalize the similarities: $\hat{\mathbf{s}}_i = \frac{\mathbf{s}_i}{\sum_j s_{ij}}$
8:     Aggregate $w_i^t$ with the models from the sampled clients:

$$w_i^{\text{merged}} \leftarrow \sum_{j=1}^{m+1} \alpha_j w_j,$$

    where $\alpha_j = \frac{n_j}{N}$ for FedAvg, and $\alpha_j = \hat{s}_j$ for FedSim.
9:     Update the merged model locally for $E$ epochs:

$$w_i^{t+1} \leftarrow w_i^{\text{merged}} - \eta \nabla_{w_i} \ell(w_i^{\text{merged}}; x_i, y_i)$$

10: **end for**
11: **return** Final model parameters for client $i$: $w_i^T$

---

# 3 Experimental setup

In this section, we describe our experimental setup for studying the effect of common similarity metrics on client identification within decentralized learning. Empirical loss and cosine similarity on gradients have been previously employed (Onoszko et al., 2021; Li et al., 2022; Listo Zec et al., 2022). Additionally, we introduce the Euclidean distance ($L^2$) between model parameters and cosine similarity between parameters (as opposed to gradients) as two additional baselines, which have not been tested before in DL for client identification. We define the similarity metrics between two clients $i$ and $j$ as follows:

- Inverse empirical loss $= \left( \sum_{(x,y) \in \mathcal{D}_j} \ell(w_i; x, y) \right)^{-1}$.

- Inverse $L^2$ distance $= \|w_i - w_j\|_2^{-1}$.

- Cosine similarity of weights $= \frac{w_i \cdot w_j}{\|w_i\| \|w_j\|}$.

- Cosine similarity of gradients $= \frac{g_i \cdot g_j}{\|g_i\| \|g_j\|}$.

For cosine similarity of gradients, we define the gradient at time $t$ as:

$$g_i^t = w_i^t - w_i^0, \tag{5}$$

where $g_i^t$ represents the vector describing the direction from the initialized weights to the current weights of the network.

## 3.1 Computational cost analysis

When selecting similarity metrics for client identification in decentralized learning, it is crucial to consider their computational overhead, especially in resource-constrained decentralized settings. In this section, we analyze the computational complexity of each of the four similarity metrics at each communication round, focusing on the operations required to compute the similarity score itself, rather than the cost of training epochs.

**Inverse empirical loss.**  To compute the inverse empirical loss as a similarity metric between client $i$ and client $j$, we evaluate client $i$'s model with parameters $w_i$ on *every data point* $(x, y)$ in client $j$'s local dataset $\mathcal{D}_j$. This necessitates performing a forward pass through the model for each data point in $\mathcal{D}_j$, resulting in a computational complexity of $\mathcal{O}(N_j \times C_f)$, where $N_j$ is the size of the dataset $\mathcal{D}_j$ and $C_f$ is the cost of a single forward pass. This approach, which deterministically evaluates the loss over the entire dataset $\mathcal{D}_j$, imposes a significant computational burden, especially with large datasets or complex models, making it less feasible in practice. While empirical loss could be stochastically estimated using a subset of $\mathcal{D}_j$ (e.g., a mini-batch), we use the full dataset to obtain a more deterministic and potentially more reliable measure of functional similarity for guiding peer selection.

**Inverse $L^2$ distance.**  The inverse Euclidean distance between model parameters $w_i$ and $w_j$ is computed directly from the parameter vectors and has a computational complexity of $\mathcal{O}(P)$, where $P$ is the number of model parameters. This calculation is deterministic and scales linearly with model size, remaining feasible even for large models.

**Cosine similarity of weights and gradients.**  Both the cosine similarity of weights and the cosine similarity of gradients, as defined by equation 5 using $g_i^t = w_i^t - w_i^0$, have a computational complexity of $\mathcal{O}(P)$, similar to the inverse $L^2$ distance. These metrics involve computing dot products and norms of vectors of size $P$, which are deterministic operations manageable even for models with a large number of parameters.

**Practical example.** To illustrate the practical implications of these computational costs, consider a moderately sized CNN with $P = 1,000,000$ parameters and a local dataset size of $N_j = 10,000$ samples per client. Assuming an idealized scenario with efficient hardware where the cost of a single forward pass $C_f$ is approximately 10 milliseconds, computing the inverse empirical loss for similarity assessment would require a total time of $N_j \times C_f = 10,000 \times 10 \text{ ms} = 100,000 \text{ ms} = 100 \text{ s}$. This calculation focuses solely on computation time and does not account for potential communication overhead in a decentralized setting.

In contrast, computing the inverse $L^2$ distance or cosine similarities involves approximately $P = 1,000,000$ basic vector operations (dot products, norms, subtractions). Assuming each operation takes approximately 1 nanosecond, the total time is roughly $P \times 1 \text{ ns} = 1,000,000 \text{ ns} = 1 \text{ ms}$. This example, while simplified, highlights the order of magnitude difference in computational cost between empirical loss and the other parameter-based similarity metrics, emphasizing the trade-offs between functional similarity measures and computational efficiency.

## 3.2 Distributional shifts in decentralized learning

When local client data generating processes are identical, meaning $P_i(x, y) = P_j(x, y) \; \forall i, j$, random communication is optimal as all clients are learning the same model. However, this assumption is often unrealistic. In practical scenarios, distribution shifts between clients are common, leading to client data heterogeneity (non-iid data). Given the variety of ways two probability distributions can differ, it is crucial to formally describe the type of shift to understand the effectiveness of decentralized learning solutions.

Consider the factorized form of the joint probability distribution:

$$P(x, y) = P(x|y)P(y) = P(y|x)P(x). \tag{6}$$

Using this factorization, we can describe some common types of distribution shifts. In this paper, we investigate the following distributional shifts in the context of decentralized learning.

1. **Covariate shift:** This occurs when $P(x)$ varies between clients while the conditional distribution $P(y|x)$ remains constant. For instance, different hospitals may use different x-ray machines, leading to variations in the input features collected, while the diagnosis based on the x-rays remains consistent across hospitals.

2. **Label shift:** This shift is present when $P(y)$ varies between clients but $P(x|y)$ is shared. An example of this scenario is mobile phone users taking photos of different types of animals, resulting in a variation of the label distribution across users.

3. **Concept shift:** This type of shift occurs when $P(y|x)$ varies but $P(x)$ stays the same. In this case, the same inputs result in different labels. This can occur when modeling user preferences or sentiment.

4. **Domain shift:** This represents a more general shift where both the conditionals $P(y|x)$ and the marginals $P(x)$ differ simultaneously across clusters. For example, consider a voice recognition system trained on data from adults speaking in quiet environments. If the system is deployed to recognize children's voices in noisy environments like playgrounds, both the input features $x$ (voice characteristics and background noise) and the relationship between these features and the labels $y$ (spoken words) change significantly, illustrating a domain shift.

Understanding these shifts is essential for the development and evaluation of decentralized learning algorithms. We evaluate the performance of similarity metrics using ERM in a decentralized setting on four datasets in the presence of distributional shifts. In all cases, we evaluate each client model $w_i$ on a test set drawn from its underlying data generating process $P_i(x, y)$. The first dataset is a synthetic one, while the three others are classic benchmark image datasets (MNIST, CIFAR-10 and Fashion-MNIST). We create different clusters of the datasets in order to simulate heterogeneous data distributions among clients. The synthetic dataset is used to study concept shift. For the benchmark datasets, we study covariate shift, label shift and domain shift. A summary of our experimental setup is shown in Table 1.

| Dataset | Shift | Model | No. of clusters | No. of clients |
|---|---|---|---|---|
| CIFAR-10 | Label | CNN | 2 | 100 |
| CIFAR-10 | Label | CNN | 5 | 100 |
| Synthetic data | Concept | Linear | 3 | 99 |
| Fashion-MNIST | Covariate | CNN | 4 | 100 |
| Fashion-MNIST + MNIST | Domain | MLP & CNN | 2 | 100 |
| CIFAR-100 | Label | Pre-trained ResNet-18 | 3 | 52 |

Table 1: Summary of our experimental setup.

### 3.3 Model selection and hyperparameter tuning

In both federated and decentralized learning environments, the process of model selection and the setting of hyperparameters are crucial yet often under-discussed elements. A common challenge in the literature is the lack of detailed reporting on hyperparameter choices in decentralized settings, which complicates the task of conducting fair and consistent methodological comparisons. Variations in results that stem from differing hyperparameter tuning practices can be mistakenly attributed to the intrinsic qualities of the algorithms themselves, thereby obscuring accurate evaluations. In practice, while some studies use a global validation set, others may employ a local validation set specific to each client, and still others might not clearly disclose their hyperparameter selection strategy.

In our investigation, we strive for a rigorous comparison across all models and similarity metrics by using a local validation set for each client. We tune all hyperparameters using these validation sets, and employ early stopping on each client locally. During communication, some clients may reach early stopping before others. When this occurs, these clients continue to broadcast their models to other clients but cease updating their own models through further training or merging. This approach ensures that our evaluation is both relevant and tailored to the specific conditions of each client's data. Additionally, we introduce a comparison against an Oracle baseline, which has access to clients with the same data distribution and only communicates with those, providing a benchmark under ideal conditions, and a Local baseline which only performs local training of the available data on a single client.

After hyperparameters were selected, each method was run three independent times, and we report the means and standard deviations across these runs. The found hyperparameters for each problem is presented in Appendix A in Table 8, together with more experimental details. For the synthetic dataset, linear models trained with SGD were used. In these experiments, 15 independent runs were made for each method. For the benchmark datasets, the models comprised neural networks with two convolutional layers followed by two linear layers. In the domain shift experiment, we also employed a single-layer MLP with ReLU activation. While these models do not represent the state-of-the-art for these tasks, they provide sufficient complexity to discern meaningful differences between the methods under study. This setup not only highlights the capabilities of the similarity metrics within a controlled experimental environment but also provides insights into their practical applicability in more realistic, decentralized scenarios.

### 3.4 Distributional shifts and datasets

**Concept shift.** We begin by studying a decentralized learning problem using linear regression in a synthetic setup. In this scenario, we assume that each cluster is defined by $y_i = \langle x_i, \theta_j^* \rangle + \varepsilon_i$, for client $i$ and cluster $j$ where $\varepsilon_i$ denotes normally distributed noise independent of the data $x$. Essentially, each cluster has a underlying data generating process defined by $\theta_j^*$ which a model for that cluster aims to learn from the data. If the $\theta_j^*$ vary significantly across clusters, merging models from different clusters would lead to suboptimal models.

To construct the clusters, we generate three distinct $\theta_j^* \in \mathbb{R}^d$, each drawn from a uniform distribution. The feature matrix $X \in \mathbb{R}^{n \times d}$ is populated by entries sampled uniformly from the range $[-10, 10]$, forming a $d$ dimensional vector with $n$ samples for each client. Finally, the response variable $y_i$ is computed as $y_i = \langle x_i, \theta_j^* \rangle$. We use mean squared error as the loss function, setting $d = 10$ and $n = 50$ for each client. We

create the clusters such that they contain an equal number of clients. This synthetic setup enables a controlled exploration of the efficacy of similarity metrics in a setting characterized by distinct, non-overlapping data distributions. This method of data generation results in a concept shift between clients – where $P(y|x)$ varies, while $P(x)$ remains constant.

**Label shift.** Using the CIFAR-10 dataset (Krizhevsky et al., 2009), we create two more realistic and complex scenarios. The first scenario divides the dataset into two clusters based on content categories: an animal cluster, comprising four labels, and a vehicle cluster, with six labels. These clusters consist of 40 and 60 clients, respectively. In the second scenario, we create five equally large clusters of 20 clients where each cluster is defined by two unique, randomly sampled labels. In this way, each cluster has a unique data generating process where $P(y)$ varies while $P(x|y)$ remains constant. We also study large scale **pre-trained models** under label shift. For this, we use the CIFAR-100 dataset (Krizhevsky et al., 2009), which we split into three distinct clusters of sizes 26, 13 and 13 based on the superclasses.

**Covariate shift.** To study covariate shift, we use the Fashion-MNIST dataset (Xiao et al., 2017). We construct four clusters, where each cluster is characterized by a specific rotation angle that defines their respective data generation process $P_k(x, y)$. This setup not only tests the impact of image rotation on model performance but also incorporates varying cluster sizes to mimic realistic client distribution: 70, 20, 5, and 5 clients per cluster. The rotation degrees set for these clusters are $0°, 10°, 180°$ and $350°$. This arrangement creates three clusters with relatively similar conditions and one notably distinct cluster – the $180°$ rotation. In this way, each cluster has a unique data generating process where $P(x)$ varies while $P(y|x)$ remains constant.

**Domain Shift.** We also explore domain shift, a common issue in the multi-task learning literature where a single model attempts to learn multiple domains (or tasks). In this setting, both $P(y|x)$ and $P(x)$ vary between clusters. We use the MNIST (LeCun et al., 1998) and Fashion-MNIST datasets, defining two clusters by partitioning these datasets into halves, each assigned to different clients.

### 3.5 FedSim: leveraging similarity metrics for weighted averaging

In our empirical studies, we observed that the traditional FedAvg, which weights updates by the number of data samples per client, can lead to catastrophic forgetting when a client's model is combined with a poorly performing model from another client due to sampling a client from a cluster with a different data distribution. To explore potential mitigations, we introduce a novel approach termed Federated Similarity Averaging (FedSim). In FedSim, each client's contribution to the global model is weighted by a similarity metric rather than merely the data count. Specifically, for a single client, the update is computed as:

$$w^{t+1} = \sum_{k=1}^{m} s_k w_k^t, \tag{7}$$

where $s_k$ represents the similarity metric for client $k$, and $w_k^t$ denotes the model parameters from client $k$ at iteration $t$. To incorporate the model updates from the client itself, we assign its weight as the maximum similarity observed among all $s_k$. Subsequently, we normalize these weights to ensure they sum to one. This methodological adjustment aims to prevent the adverse effects observed with traditional federated averaging by prioritizing contributions from clients that have more aligned optimization problems.

**Limitations of FedAvg under distribution shift** The FedAvg aggregation does not yield an unbiased estimate of the target distribution $P_k$ that a client associated with $P_k$ aims to learn when merging models trained on other distribution $P_i \neq P_k$. Specifically, the expected value of the aggregated model using FedAvg is

$$\mathbb{E}[w^{t+1}] = \sum_{i=1}^{m} \frac{N_i}{N} \mathbb{E}[w_i^t] = \sum_{i=1}^{m} \frac{N_i}{N} w_i^\star \neq w_k^\star, \tag{8}$$

where $N_i$ is the number of samples for client $i$ and $N = \sum_i N_i$, $w_i^\star$ is the optimal model parameters for distribution $P_i$, and $w_k^\star$ the optimal model parameters for distribution $P_k$. The last inequality in equation 8 arises because the combination of the optimal $w_i^\star$ from **other distributions** $P_i$ for $i = 1, \dots, m$ does not, in general, equal $w_k^\star$.

To address this limitation, we propose using similarity-based averaging. Under the assumption that the target distribution $P_k$ can be expressed as a weighted combination of the sampled client distributions $P_i$, we can show that similarity-based averaging provides and unbiased estimate of the optimal parameters for $P_k$.

**Assumption 1.** *The target $P_k(X, Y)$ can be expressed as a weighted combination of the sampled client distributions, i.e. $P_k(X, Y) = \sum_{i=1}^{m} s_i P_i(X, Y)$, where $\mathbf{s} = (s_1, s_2, \dots, s_m)$ is a vector of weights satisfying $s_i \geq 0$ and $\sum_{i=1}^{m} s_i = 1$.*

Under Assumption 1, we have the following proposition.

**Proposition 1** (Unbiased estimator)**.** *Consider a single round $t$ in the batch stochastic gradient setting with learning rate $\eta$. Let each sampled client $i \in [m]$ compute its local update $w_{i,t} = w_t - \eta \nabla_w \hat{R}_i(w_t)$, where $\hat{R}_i(w_t)$ is the empirical risk on client $i$. Then, the aggregated update using the similarity weights $\mathbf{s}$ is $w_{t+1} = \sum_{i=1}^{m} s_i w_{i,t}$. Under Assumption 1, the expected aggregated update satisfies*

$$\mathbb{E}[w_{t+1} \mid w_t] = \mathbb{E}[w_{t+1}^{P_k} \mid w_t],$$

*where $w_{t+1}^{P_k} = w_t - \eta \nabla_w \hat{R}_{P_k}(w_t)$ is the batch SGD update that would be obtained using a sample from the target distribution $P_k$.*

*Proof sketch.* Under Assumption 1, the expected gradient of the aggregated update can be written as

$$
\begin{aligned}
\mathbb{E}[w_{t+1} \mid w_t] &= \sum_{i=1}^{m} s_i \mathbb{E}[w_{i,t} \mid w_t] = \sum_{i=1}^{m} s_i \left( w_t - \eta \mathbb{E}[\nabla_w \hat{R}_i(w_t)] \right) = \\
w_t &- \eta \sum_{i=1}^{m} s_i \mathbb{E}[\nabla_w \hat{R}_i(w_t)] = w_t - \eta \mathbb{E}_{(X,Y) \sim P_k}[\nabla_w \ell(w_t; X, Y)] = w_{t+1}^{P_k}.
\end{aligned}
\tag{9}
$$

$\square$

Under this proposition, the expected value of the model update in FedSim with *perfect* similarity scores $s_i$ is

$$\mathbb{E}[w^{t+1}] = \sum_{i=1}^{m} s_i \mathbb{E}[w_i^t] = \sum_{i=1}^{m} s_i w_i^\star. \tag{10}$$

Thus, similarity-based averaging with perfect similarity scores provides an unbiased estimate of the optimal model parameters for the target distribution. We present results from our experimental work analyzing common similarity metrics used in decentralized learning in Section 4, both for *sampling clients* and for *aggregation using FedSim*.

## 4 Results

In this section, we illustrate the effectiveness of the similarity metrics. Unless otherwise stated, *all results are means over three independent runs*. The main results are presented in Table 2.

### 4.1 Concept shift

**Synthetic experiment.** In the synthetic experiment, we generated three distinct clusters where the conditional distribution $P(y|x)$ varies between clusters while the marginal $P(x)$ remains constant. *In these experiments, all reported results are means over 15 independent runs.* The findings, illustrated in Figure 3a, demonstrate a significant improvement in performance using FedSim compared to FedAvg, especially

Table 2: Performance comparison across methods and tasks on test sets. Mean squared error (MSE) is used for the synthetic experiment, while test accuracy is used for all other tasks. The reported values represent the averages across all clusters for each method and experiment. Best performing method (excluding the Oracle) is denoted with **bold text**.

| | Synthetic ($\downarrow$) | CIFAR-10 ($\uparrow$) | CIFAR-10 ($\uparrow$) | F/MNIST ($\uparrow$) | FMNIST ($\uparrow$) | CIFAR-100 ($\uparrow$) |
|---|---|---|---|---|---|---|
| No. of clusters | 3 | 2 | 5 | 2 | 4 | 3 |
| Shift | Concept | Label | Label | Domain | Covariate | Label |
| $L^2$ FEDAVG | $21.10 \pm 2.51$ | $59.44 \pm 0.66$ | $84.04 \pm 0.63$ | $90.49 \pm 0.12$ | $84.17 \pm 0.21$ | $\mathbf{55.53 \pm 0.57}$ |
| $L^2$ FEDSIM | $10.85 \pm 0.33$ | $57.63 \pm 1.14$ | $84.06 \pm 0.31$ | $90.80 \pm 0.11$ | $84.16 \pm 0.01$ | - |
| Inv loss FEDAVG | $31.69 \pm 3.23$ | $\mathbf{61.62 \pm 0.83}$ | $84.94 \pm 0.53$ | $89.83 \pm 0.22$ | $84.83 \pm 0.14$ | $54.23 \pm 0.21$ |
| Inv loss FEDSIM | $14.82 \pm 0.33$ | $61.08 \pm 1.15$ | $85.80 \pm 0.20$ | $90.67 \pm 0.07$ | $\mathbf{85.08 \pm 0.26}$ | - |
| Cos grad FEDAVG | $10.32 \pm 0.23$ | $58.81 \pm 0.88$ | $\mathbf{86.22 \pm 0.17}$ | $91.14 \pm 0.17$ | $84.87 \pm 0.29$ | $53.87 \pm 0.72$ |
| Cos grad FEDSIM | $10.30 \pm 0.25$ | $58.13 \pm 0.75$ | $86.00 \pm 0.32$ | $90.72 \pm 0.24$ | $83.32 \pm 0.58$ | - |
| Cos weight FEDAVG | $10.34 \pm 0.28$ | $59.14 \pm 1.62$ | $85.85 \pm 0.20$ | $91.03 \pm 0.23$ | $85.04 \pm 0.13$ | $54.54 \pm 0.04$ |
| Cos weight FEDSIM | $\mathbf{10.30 \pm 0.21}$ | $59.66 \pm 1.02$ | $85.66 \pm 0.36$ | $90.62 \pm 0.17$ | $84.82 \pm 0.42$ | - |
| Random | $1494.84 \pm 1973$ | $59.71 \pm 1.13$ | $82.73 \pm 0.33$ | $90.53 \pm 0.05$ | $83.70 \pm 0.12$ | $54.88 \pm 0.27$ |
| Local | $30.26 \pm 1.73$ | $37.77 \pm 0.24$ | $77.08 \pm 0.98$ | $82.22 \pm 0.30$ | $76.26 \pm 0.12$ | $29.49 \pm 0.03$ |
| Oracle | $9.43 \pm 0.33$ | $62.25 \pm 0.47$ | $87.36 \pm 0.20$ | $91.62 \pm 0.08$ | $85.55 \pm 0.11$ | $56.37 \pm 0.17$ |

for the inverse loss and $L^2$ metrics. This highlights the challenges associated with merging models from different clusters for FedAvg, as further illustrated in Table 2, which shows that random communication fails to address the problem effectively. Since $P(y|x)$ differs between clusters, a single model fails to adequately represent all the data. In this scenario, the Oracle method, represented by the green area in Figure 3a, performs the best. In contrast, the inverse loss metric underperforms relative to other metrics, struggling to identify the correct clusters due to noisy sampling, where clients frequently sample incorrect peers, leading to catastrophic forgetting during model merging. Notably, cosine similarities on gradients and weights approach optimal performance with minimal standard deviation across runs. Figure 2a presents a heatmap of client communication, clearly indicating that the inverse loss and $L^2$ metrics exhibit more noise compared to cosine similarity metrics.

We also conducted an experiment to examine the effect of training sample size on the similarity metrics. As shown in Figure 3b, we repeated the same experiment with an increased number of samples for each method. The results indicate that as the number of training samples increases, the performance gap for inverse loss and $L^2$ distance metrics narrows. However, these metrics never match the performance of cosine similarity on gradients and weights, which achieve a low test loss even with fewer training samples.

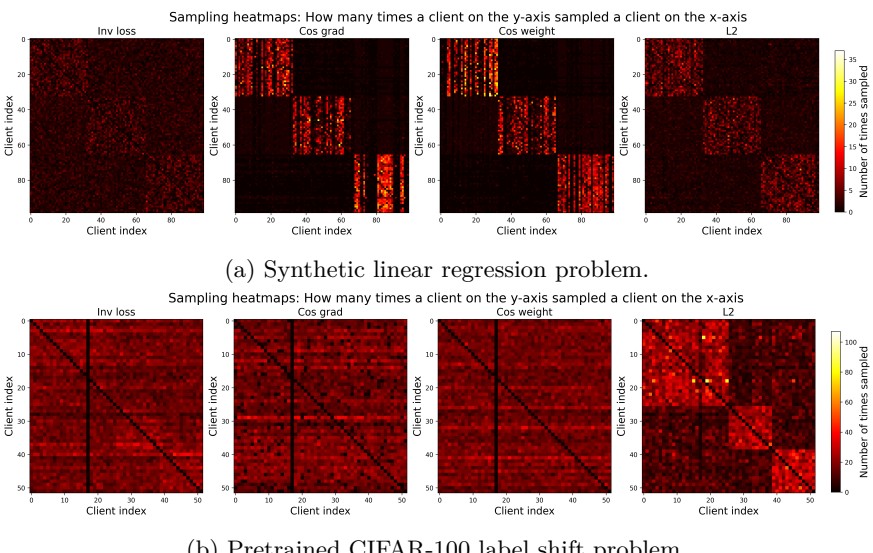

(a) Synthetic linear regression problem.

(b) Pretrained CIFAR-100 label shift problem.

Figure 2: Heatmaps of client communication, indicating how often client $x$ communicated with client $y$ for the four different similarity metrics.

## 4.2 Domain shift

**FashionMNIST and MNIST.** We further investigate the impact of similarity metrics on domain shift in a nonconvex optimization problem using neural networks. In this scenario, we form two clusters: one consisting of clients with data from the MNIST dataset, and the other with data from the Fashion-MNIST dataset. Experiments were conducted using both a Multi-Layer Perceptron (MLP) and a Convolutional Neural Network (CNN) to assess the influence of model capacity on concept shift.

As illustrated in Figure 4a, there is a noticeable performance gap between random communication and the Oracle method (red and green, respectively). In this setup, FedSim negatively impacts all similarity metrics except for the $L^2$ distance. This suggests that when the similarity metrics effectively separate the clusters, FedSim may hinder learning by causing clients to underweight models from other clients within the same cluster. Similar results are observed with the CNN, as shown in Figure 4b. Notably, inverse loss performs significantly worse than the other metrics in this context, indicating that it samples incorrect peers too often. Incorrect merging lowers performance of the local client models, which in turn makes empirical loss a worse similarity metric. In both the MLP and CNN settings, cosine similarity on gradients or weights outperforms the other metrics, exhibiting relatively low variance in test accuracies. This indicates that cosine similarity is a robust choice for decentralized learning in environments with distributional shifts, applicable to both model complexities.

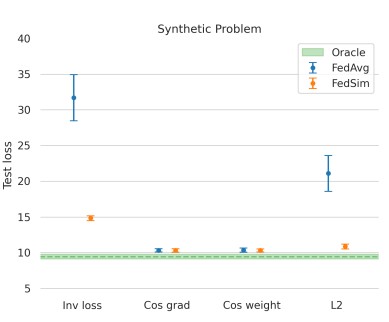
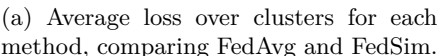

(a) Average loss over clusters for each method, comparing FedAvg and FedSim.

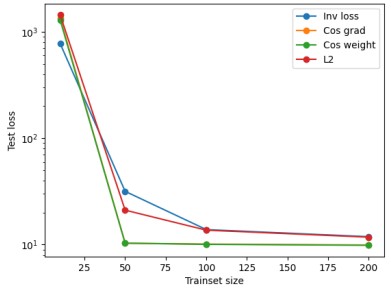

(b) Average loss over clusters as a function of trainset size for all methods (FedAvg). The orange line depicting cos grad is hidden as it follows the green line.

Figure 3: Test loss performance for all methods on the linear regression problem with concept shift.

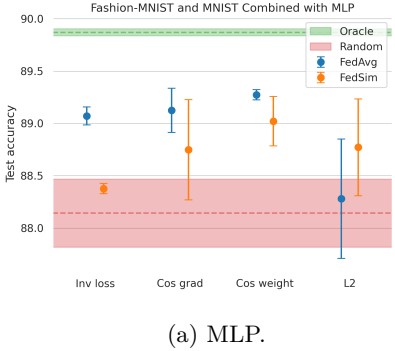

(a) MLP.

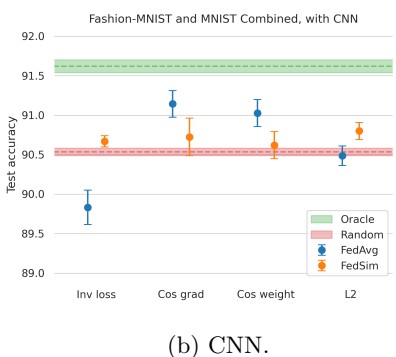

(b) CNN.

Figure 4: Test accuracy for all methods on the domain shift problem, where half of clients have data from the MNIST dataset and the other half have data from the Fashion-MNIST dataset. Results for two different model architectures: (a) an MLP and (b) a CNN.

### 4.3 Covariate shift

**Fashion-MNIST.** For the covariate shift experiments, we observe that all similarity metrics outperform random communication, except for cosine similarity on gradients when using FedSim. This is illustrated in Figure 5a and Table 2. FedAvg consistently demonstrates more stable performance across different metrics compared to FedSim, suggesting that the similarity metrics effectively separate the clusters. Among the metrics, inverse loss, cosine similarity on gradients, and cosine similarity on weights exhibit relatively comparable performance. However, the $L^2$ distance metric underperforms relative to the other metrics, despite still achieving better results than random communication. These findings underscore the effectiveness of using FedAvg with appropriate similarity metrics to handle covariate shifts in decentralized learning.

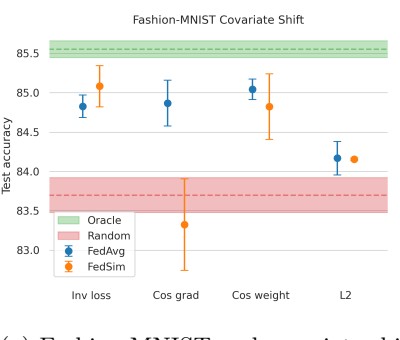

(a) Fashion-MNIST and covariate shift.

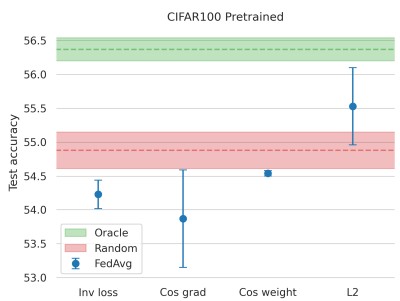
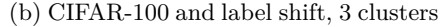

(b) CIFAR-100 and label shift, 3 clusters.

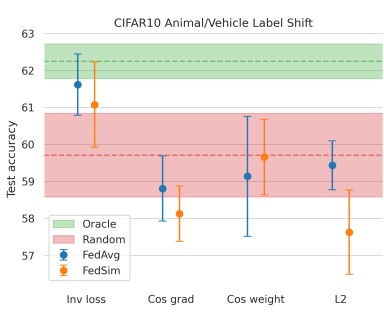

(c) CIFAR-10 and label shift, 2 clusters.

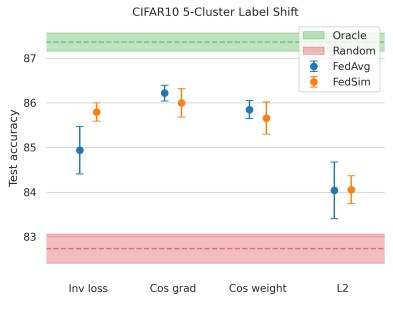

(d) CIFAR-10 and label shift, 5 clusters

Figure 5: Test accuracy for all methods on different problems: Fashion-MNIST covariate shift (a), pretrained CIFAR-100 label shift (b), and the CIFAR-10 label shifts (c,d).

### 4.4 Label shift

**CIFAR-10, 2 clusters.** In the CIFAR-10 experiment with an animal cluster and a vehicle cluster, most metrics do not exceed the performance of random communication, presented in Figure 5c and Table 2. Additionally, the gap between random communication and the Oracle method is small, suggesting that this specific label shift relatively straightforward for random communication to manage. Nevertheless, we observe a performance increase for the inverse loss metric in this setting as it outperforms random communication and the other metrics.

**CIFAR-10, 5 clusters**. In the scenario with five clusters, where each cluster is defined by its two unique labels, we observe a larger performance gain across all metrics. This is presented in Figure 5d and Table 2. Here, the label shift is more pronounced as $P(y)$ differs more significantly between the clusters. This means that merging models with wrong peers has a larger negative impact on performance. This is illustrated by the fact that random communication achieves even worse performance compared to the Oracle method in this setup, as compared to the previous setup of two clusters. The cosine similarity metrics, however, demonstrate robust performance with low variance across runs. In this setting, inverse loss does not match

the performance of the cosine similarity metrics. Meanwhile, using FedSim improves the performance of inverse loss, indicating that inverse loss is able to identify clusters, albeit noisily. The $L^2$ metric performs worse than all other metrics but still manages to outperform random communication.

**CIFAR-100, 3 clusters, pre-trained ResNet-18.** Previous research in federated learning has shown that starting with a pre-trained model can mitigate some of the adverse impacts of label shift (Nguyen et al., 2022). In our experiments, we observed that this approach makes the Random method relatively stronger. As depicted in Figure 5b, three out of the four similarity metrics did not outperform the Random method when using a pre-trained model. For the inverse loss metric, the pre-trained model's robustness results in uniformly low loss across all clients, causing all clients to appear very similar. In the case of cosine similarity on gradients and weights, the similarity scores between clients were also too uniform, making it challenging to detect significant cluster structures. Notably, the $L^2$ metric was the only method capable of accurately clustering clients and outperforming random communication in this context. This is further illustrated in Figure 2b, which presents communication heatmaps and highlights the difficulties these methods encounter in correctly identifying clusters. In these experiments, we did not evaluate FedSim with pre-trained models due to the substantial computational resources required for training large-scale peer-to-peer systems with large models. Future work will address this limitation by exploring both FedSim and other aggregation methods for decentralized learning.

## 5   Conclusions

We have conducted a thorough empirical study on the effect of similarity metrics in decentralized learning for client identification. Our findings highlight the strengths and weaknesses of various similarity metrics, which are dependent on the problem type and the nature of the distributional shift. Based on our results, we recommend that researchers and practitioners carefully explore similarity metrics in decentralized learning to identify the most suitable one for their specific problem. Our study indicates that the effectiveness of similarity metrics varies depending on the specific conditions and types of distributional shifts encountered. Our key conclusions are as follow:

i **The $L^2$ distance metric:** The $L^2$ metric, although outperforming random communication most of the time, is the weakest of the similarity metrics we studied. We hypothesize that this is due to the metric being sensitive to the scaling of parameters and highly susceptible to permutations of neurons or layers, making it less reliable when comparing models with different weight configurations but similar functional behavior.

ii **Inverse empirical loss:** The inverse empirical loss provides a direct functional measure of similarity. However, it is highly dependent on the specific training data used, with its reliability being contingent on the number of samples and their quality. Clients with inherently more difficult tasks or poorer initial models can have higher loss values, which can skew the inverse loss and mislead the merging process. If the sample size is large, and the local models are able to solve the local problem well, this metric is effective at identify beneficial peers for model merging.

iii **Cosine similarity:** Our results indicate that cosine similarity on model parameters often outperforms other metrics, particularly the $L^2$ method. This is likely due to the fact that cosine similarity is less sensitive to scaling compared to $L^2$ distance and captures the similarity in the direction of weights rather than their exact values, making it more robust to permutations. Similarly, cosine similarity on gradients provides valuable insights into the learning dynamics between models, beyond their static states. Our findings suggest that cosine similarity metrics are relatively robust across various scenarios.

iv **Concept shift:** Clustering of clients in decentralized learning is especially useful in the presence of strong concept shift, where $P(y|x)$ varies across clusters while $P(x)$ remains constant. In these cases, random communication cannot solve the optimization problem, necessitating accurate cluster identification.

v **Random communication:** Despite various distributional shifts our results demonstrate that with proper and robust model selection, random communication remains a strong baseline.

vi **Pre-training:** Pre-training can enhance the performance of random communication in scenarios with label shift. However, the use of a robust pre-trained model tends to standardize similarity scores, making it challenging for various metrics to effectively identify distinct clusters.

vii **Federated Similarity Averaging (FedSim):** We introduced a new aggregation scheme, FedSim. While it is not a silver bullet, FedSim can enhance the performance of FedAvg, particularly in scenarios where averaging with incorrect models incurs high costs, such as in the presence of strong concept shift between clusters (Figure 3a).

Overall, our study underscores the importance of selecting appropriate similarity metrics and aggregation strategies based on the specific context and challenges of decentralized learning tasks.

## 6 Future work

There are several promising directions for future research, both theoretical and empirical. Developing theoretical frameworks for measuring similarity between private datasets could provide better insights into the optimal selection of similarity metrics.

The research area of decentralized learning on non-iid data is closely related to domain generalization and transfer learning. These fields involve understanding when it is possible to jointly train deep learning models across different probability distributions and determining the extent to which these models can extrapolate to new data. Bridging the gap between these areas of research is an interesting direction.

Additionally, investigating the intersection of similarity metrics and privacy is a compelling direction. Beyond model-based similarity metrics like inverse empirical loss, Euclidean distance, and cosine similarity – which assess similarity by comparing clients' model weights – *optimal transport* theory offers a framework for measuring similarity based on clients' data distributions (Peyré et al., 2019). The *Wasserstein distance*, a fundamental concept in optimal transport, quantifies the minimal cost of transforming one probability distribution into another. While the Wasserstein distance provides a nuanced and theoretically sound measure of data distribution similarity, its practical application is hindered by computational complexity. Computing the exact Wasserstein distance for high-dimensional empirical distributions involves solving optimization problems with significant resource demands. Moreover, using the Wasserstein distance in decentralized learning introduces considerable privacy concerns. Since this metric relies on detailed statistical information or even direct access to empirical data distributions, it risks exposing sensitive information, thereby conflicting with the privacy-preserving goals of decentralized systems. Future work should focus on addressing these computational and privacy challenges associated with advanced similarity metrics like the Wasserstein distance. Developing efficient approximations or privacy-enhancing techniques tailored to these metrics could significantly enhance their utility in decentralized learning, fostering more robust, generalizable, and privacy-conscious machine learning models.

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

# A Appendix

## A.1 Hyperparameters and experimental details

In our study, we aimed for a rigorous and fair evaluation where we performed a grid search over hyperparameters to optimize them for each experimental setup. We ensured that no selected hyperparameter was at the boundary of the grid. If this occurred, we expanded the grid size accordingly. All tuning was carried out using local validation sets for each client. Each experiment was independently run three times (fifteen times for the synthetic problem) with the optimized hyperparameters.

Table 8 presents the selected hyperparameters and other experimental details. Figure 8 illustrates the impact of $\tau$ in equation 4 on performance. Calibrating $\tau$ appropriately is crucial for the effectiveness of similarity-based clustering, and the optimal value of $\tau$ varies across different problems and methods.

## A.2 Additional experimental results

In this section we present additional results. In Table 3, the results for the synthetic linear regression problem is presented for each cluster. Table 4 shows the accuracies for each cluster in the domain shift problem. 5 presents the results for all clusters in the covariate shift experiment with Fashion-MNIST. Table 6 and 7 summarizes the results on the label shift experiments for CIFAR-10.

In Figure 6 we observe how the number of sampled clients affect test accuracy in the two-cluster (animal/vehicles) CIFAR-10 experiment. In this experiment we have 200 training samples in each client, and a total of 100 clients. Sampling a lot of clients start give diminishing returns, but increases computational cost – especially for inverse loss that has to perform forward passes on all sampled clients. Therefore it is important to tune this hyperparameter to ensure a good trade-off between performance and computation.

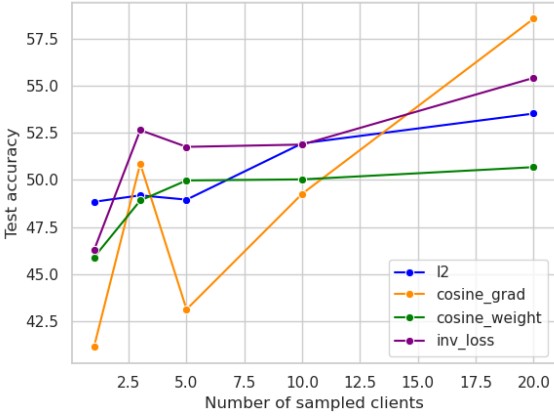

Figure 6: Test accuracy vs number of sampled clients for all similarity metrics.

In Figure 7, heatmaps over client communication for the CIFAR-10 and Fashion-MNIST are presented.

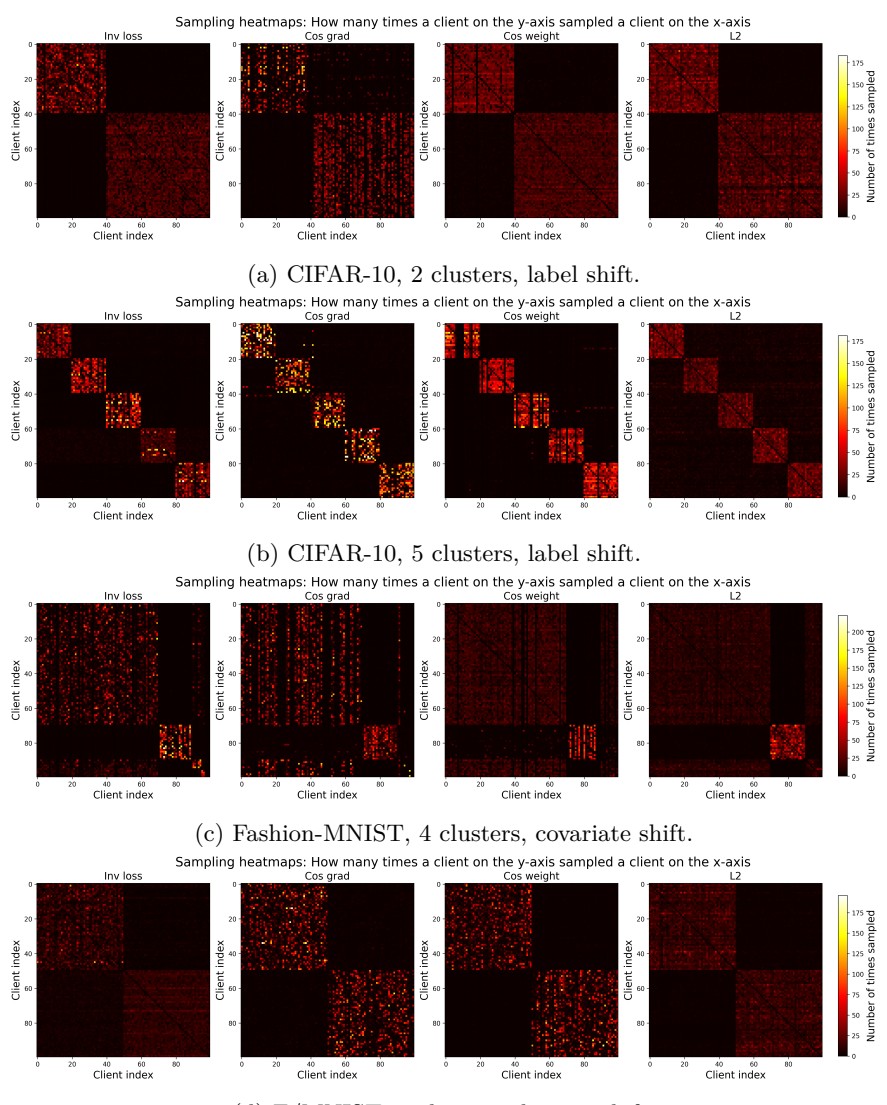

(a) CIFAR-10, 2 clusters, label shift.

(b) CIFAR-10, 5 clusters, label shift.

(c) Fashion-MNIST, 4 clusters, covariate shift.

(d) F/MNIST, 2 clusters, domain shift.

Figure 7: Heatmaps of client communication, indicating how often client $x$ communicated with client $y$ for the four different similarity metrics.

| Baselines | Cluster 1 | | Cluster 2 | | Cluster 3 | | Mean | |
|---|---|---|---|---|---|---|---|---|
| | FedAvg | FedSim | FedAvg | FedSim | FedAvg | FedSim | FedAvg | FedSim |
| Random | 1196.04 | - | 1251.50 | - | 2036.90 | - | 1494.84 | - |
| Oracle | 9.54 | - | 9.29 | - | 9.46 | - | 9.43 | - |
| Inv loss | 32.91 | 14.87 | 30.40 | 14.58 | 31.77 | 15.01 | 31.69 | 14.82 |
| Cos grad | 10.19 | 10.40 | 10.32 | 10.22 | 10.44 | 10.29 | 10.32 | **10.30** |
| Cos weight | 10.29 | 10.37 | 10.25 | 10.41 | 10.48 | 10.13 | 10.34 | **10.30** |
| $L^2$ | 21.97 | 10.85 | 19.77 | 10.92 | 21.56 | 10.78 | 21.10 | 10.85 |
| Local | 28.76 | - | 26.80 | - | 35.22 | - | 30.26 | - |

Table 3: Test loss for each method on the synthetic linear regression problem.

| Baselines | Cluster 1 | | Cluster 2 | | Mean | |
|---|---|---|---|---|---|---|
| | FedAvg | FedSim | FedAvg | FedSim | FedAvg | FedSim |
| Random | 97.01 | - | 84.05 | - | 90.53 | - |
| Oracle | 97.68 | - | 85.56 | - | 91.62 | - |
| Inv loss | 97.11 | 96.96 | 82.55 | 84.38 | 89.83 | 90.67 |
| Cos grad | 97.29 | 96.91 | 84.99 | 84.54 | 91.14 | 90.72 |
| Cos weight | 97.22 | 96.72 | 84.83 | 84.52 | 91.03 | 90.62 |
| $L^2$ | 97.05 | 97.04 | 83.92 | 84.56 | 90.49 | 90.80 |
| Local | 89.29 | - | 75.15 | - | 82.22 | - |

(a) CNN

| Baselines | Cluster 1 | | Cluster 2 | | Mean | |
|---|---|---|---|---|---|---|
| | FedAvg | FedSim | FedAvg | FedSim | FedAvg | FedSim |
| Random | 93.66 | - | 82.63 | - | 88.14 | - |
| Oracle | 95.38 | - | 84.36 | - | 89.87 | - |
| Inv loss | 94.19 | 94.73 | 83.96 | 82.02 | 89.07 | 88.38 |
| Cos grad | 94.79 | 94.54 | 83.46 | 82.96 | 89.13 | 88.75 |
| Cos weight | 95.15 | 95.18 | 83.39 | 82.86 | 89.27 | 89.02 |
| $L^2$ | 94.83 | 95.32 | 81.73 | 82.22 | 88.28 | 88.77 |
| Local | 86.07 | - | 76.34 | - | 81.20 | - |

(b) MLP

Table 4: Test accuracy comparison of methods under domain shift with two clusters (MNIST and Fashion-MNIST).

| Baselines | Cluster 1 | | Cluster 2 | | Cluster 3 | | Cluster 4 | | Mean | |
|---|---|---|---|---|---|---|---|---|---|---|
| | FedAvg | FedSim | FedAvg | FedSim | FedAvg | FedSim | FedAvg | FedSim | FedAvg | FedSim |
| Random | 85.11 | - | 79.53 | - | 82.36 | - | 81.89 | - | 83.70 | - |
| Oracle | 86.11 | - | 85.19 | - | 82.15 | - | 82.60 | - | 85.55 | - |
| Inv loss | 85.49 | 85.51 | 83.73 | 84.72 | 82.44 | 82.64 | 82.31 | 83.02 | 84.83 | **85.08** |
| Cos grad | 85.61 | 85.32 | 83.68 | 77.02 | 82.57 | 82.11 | 81.59 | 81.85 | 84.87 | 83.32 |
| Cos weight | 85.81 | 85.24 | 83.59 | 84.67 | 82.74 | 82.34 | 82.35 | 82.13 | **85.04** | 84.82 |
| $L^2$ | 85.54 | 85.37 | 80.17 | 80.87 | 82.66 | 82.21 | 82.43 | 82.26 | 84.17 | 84.16 |
| Local | 76.36 | - | 75.86 | - | 76.23 | - | 76.47 | - | 76.26 | - |

Table 5: Comparison of baselines across clusters and methods. Fashion-MNIST rotation

| Baselines | Cluster 1 | | Cluster 2 | | Mean | |
|---|---|---|---|---|---|---|
| | FedAvg | FedSim | FedAvg | FedSim | FedAvg | FedSim |
| Random | 52.88 | - | 69.94 | - | 59.71 | - |
| Oracle | 54.39 | - | 74.05 | - | 62.25 | - |
| Inv loss | 53.97 | 53.77 | 73.10 | 72.05 | **61.62** | 61.08 |
| Cos grad | 52.28 | 51.06 | 68.62 | 68.73 | 58.81 | 58.13 |
| Cos weight | 51.96 | 52.02 | 69.90 | 71.11 | 59.14 | 59.66 |
| $L^2$ | 52.14 | 50.62 | 70.39 | 68.15 | 59.44 | 57.63 |
| Local | 30.14 | - | 49.21 | - | 37.77 | - |

Table 6: Test accuracy comparison of baselines across clusters and methods. Cifar-10, 2 clusters.

| Baselines | Cluster 1 | | Cluster 2 | | Cluster 3 | | Cluster 4 | | Cluster 5 | | Mean | |
|---|---|---|---|---|---|---|---|---|---|---|---|---|
| | FedAvg | FedSim | FedAvg | FedSim | FedAvg | FedSim | FedAvg | FedSim | FedAvg | FedSim | FedAvg | FedSim |
| Random | 87.34 | - | 73.99 | - | 79.13 | - | 88.76 | - | 84.46 | - | 82.73 | - |
| Oracle | 91.66 | - | 78.64 | - | 83.25 | - | 92.2 | - | 91.06 | - | 87.36 | - |
| Inv loss | 88.1 | 90.86 | 77.83 | 76.77 | 82.14 | 81.36 | 88.37 | 91.23 | 88.26 | 88.77 | 84.94 | 85.8 |
| Cos grad | 91.3 | 90.86 | 77.21 | 77.06 | 81.91 | 82.38 | 91.44 | 91.11 | 89.23 | 88.59 | **86.22** | 86.0 |
| Cos weight | 91.01 | 91.23 | 77.46 | 76.51 | 81.16 | 81.45 | 91.42 | 91.5 | 88.19 | 87.6 | 85.85 | 85.66 |
| $L^2$ | 89.29 | 89.21 | 74.79 | 75.3 | 79.85 | 79.67 | 89.93 | 90.02 | 86.35 | 86.07 | 84.04 | 84.06 |
| Local | 82.31 | - | 66.93 | - | 74.07 | - | 83.84 | - | 78.25 | - | 77.08 | - |

Table 7: Test accuracy comparison of baselines across clusters and methods. Cifar-10, 5 clusters.

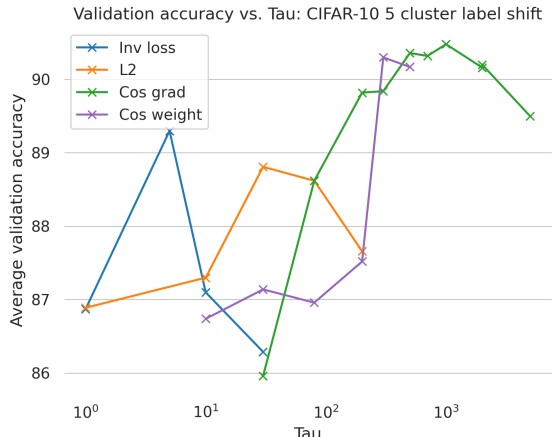

(a) CIFAR-10, 5 clusters, label shift (FedAvg).

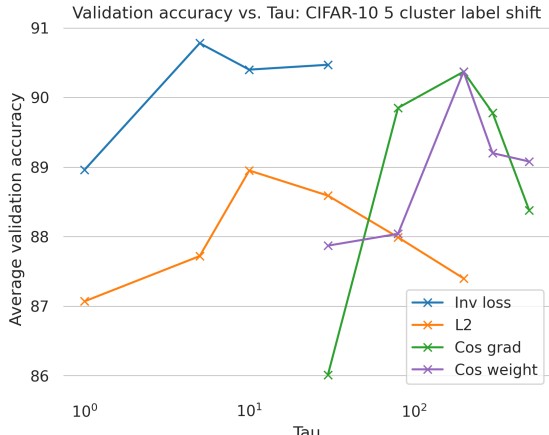

(b) CIFAR-10, 5 clusters, label shift (FedSim).

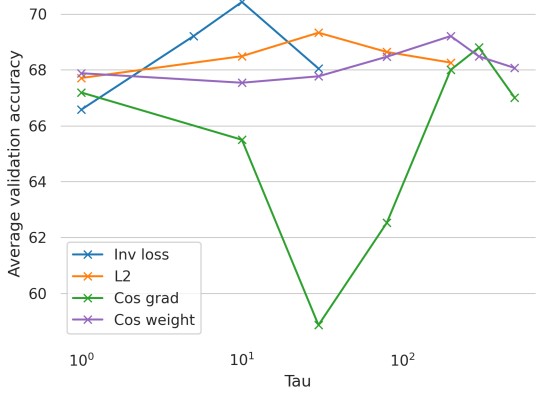

(c) CIFAR-10, 2 clusters, label shift (FedAvg).

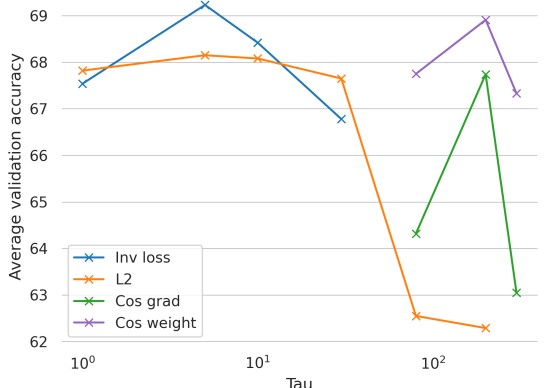

(d) CIFAR-10, 2 clusters, label shift (FedSim).

Figure 8: Sensitivity analysis of how the choice of $\tau$ effects performance for the different methods.

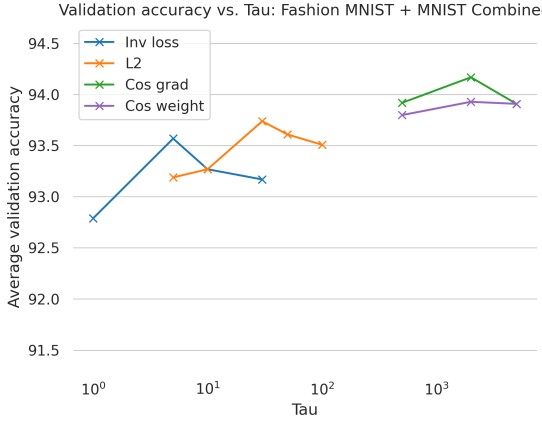

(a) F/MNIST, 2 clusters, domain shift (FedAvg).

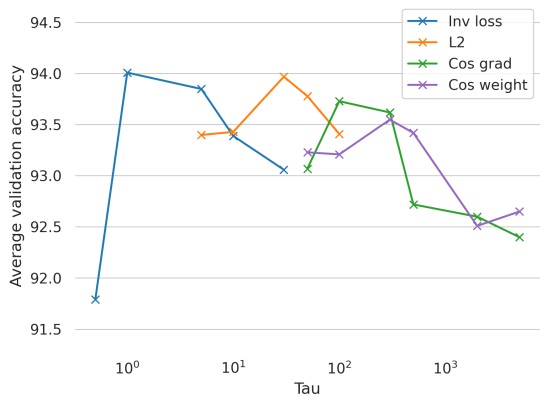

(b) CIFAR-10, 2 clusters, domain shift (FedSim).

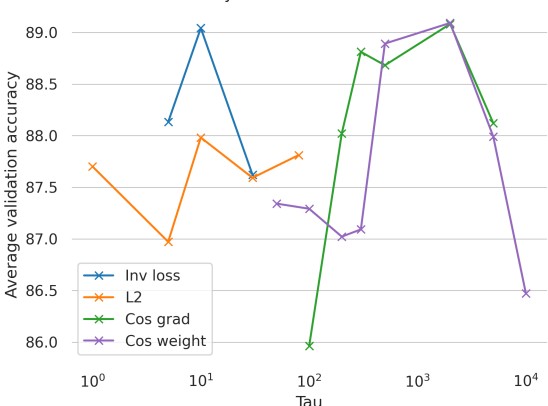

(c) Fashion-MNIST, 4 clusters, covariate shift (FedAvg).

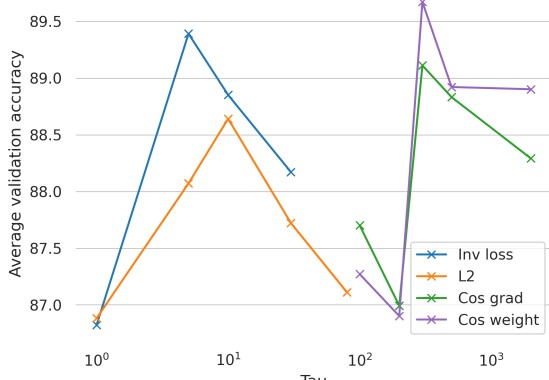

(d) Fashion-MNIST, 4 clusters, covariate shift (FedSim).

Figure 9: Sensitivity analysis of how the choice of $\tau$ effects performance for the different methods.

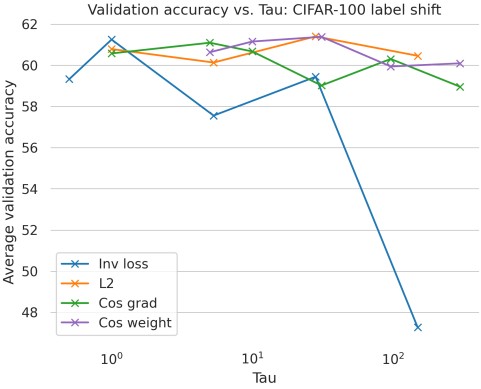

Figure 10: Sensitivity analysis of how the choice of $\tau$ effects performance for the different methods on the pre-trained CIFAR-100 experiment.

| Hyperparameters | Experiment name | | | | | |
|---|---|---|---|---|---|---|
| | CIFAR-10 animal/ vehicles label shift | CIFAR-10 5 cluster label shift | Synthetic problem | Fashion-MNIST covariate shift | Fashion-MNIST and MNIST combined | CIFAR-100 label shift |
| Learning rate | 0.001 | 0.001 | 0.003/0.008 | 0.0003 | 0.001 | 0.0001 |
| No comm learning rate | 0.0001 | 0.0001 | 0.008 | $5 \cdot 10^{-5}$ | $5 \cdot 10^{-5}$ | 0.0001 |
| No. of local epochs | 1 | 1 | 1 | 1 | 1 | 1 |
| No. of rounds | 300 | 300 | 50 | 300 | 300 | 270 |
| No. of clients | 100 | 100 | 99 | 100 | 100 | 52 |
| No. of clients per cluster | 40/60 | 20/20/20/20/20 | 33/33/33 | 70/20/5/5 | 50/50 | 26/13/13 |
| Train data size per client | 400 | 400 | 50 | 500 | 400 | 400 |
| Validation data size per client | 100 | 100 | 100 | 100 | 100 | 100 |
| Early stopping rounds | 50 | 50 | 50 | 50 | 50 | 50 |
| No. of neighbors sampled | 5 | 5 | 5 | 4 | 5 | 5 |
| No. of labels | 10 | 10 | na | 10 | 20 | 100 |
| No. of runs per similarity | 3 | 3 | 15 | 3 | 3 | 3 |
| Optimizer | Adam | Adam | Adam | Adam | Adam | Adam |
| Values used for $\tau$ | | | | | | |
| Inv loss, **FedAvg** | 10 | 5 | 10000 | 10 | 1 | 1 |
| Cos grad, **FedAvg** | 300 | 1000 | 140 | 2000 | 1 | 5 |
| Cos weight, **FedAvg** | 200 | 300 | 140 | 2000 | 5 | 30 |
| L2, **FedAvg** | 30 | 30 | 19 | 10 | 1 | 30 |
| Inv loss, **FedSim** | 5 | 5 | 5000 | 5 | 1 | na |
| Cos grad, **FedSim** | 200 | 200 | 140 | 300 | 5 | na |
| Cos weight, **FedSim** | 200 | 200 | 140 | 300 | 3 | na |
| L2, **FedSim** | 5 | 10 | 19 | 30 | 30 | na |

Table 8: Summary of all experiment settings used, as well as the found optimal hyperparameters for each experiment.

