# OpenReview forum: "On the effects of similarity metrics in decentralized deep learning under distribution shift"
_TMLR — Accepted by TMLR_

### Review · Reviewer_GnUc · 2024-08-01

**Summary Of Contributions:**

This work is an empirical study using different distance metrics to identify similar clients for decentralized learning in non-IID data. The study investigates 4 metrics under 4 distribution shifts on 4 basic ML and a synthetic dataset distributed in a way common to literature.

The studied experiments have 2 steps: first, client selection: the clients select peers to help them build a better model. Secondly, the model updates from the peers are weighted. The client selection is based on the 4 similarity metrics. The weighting is either simple FedAvg (weighted by the dataset size) or FedSim, a weighted averaging method proposed by the study that considers the similarity.

The experimental results in summary: the choice of the metric is important. The effect of the new averaging technique is marginal.

**Audience:**

Yes

**Broader Impact Concerns:**

I do not have concerns

**Claims And Evidence:**

No

**Requested Changes:**

1. Losses: 4 basic losses were used, I would recommend including in the discussion and the study optimal transport and Wasserstein distance as well.
2. Clustering: hierarchical clustering is well-studied in Federated Learning eg. [1]. The current study's literature review also mentions works for client selection. I recommend implementing at least one of these and investigating the effect of changing the distance metric behind it.
3. If someone studies non-IID data, I recommend using at least one more realistically decentralized dataset. See for example [2]
4. The choice of label pairs in CIFAR-100 impacts the experiment highly, it shouldn't be random, see [3].
5. The difference between the results in Table 2 is very small, I recommend adding the standard deviation of the 3 (15) runs.
6. Table 2 results should be in order with the subsections of the 4. Results section.
7. I think adding a discussion about the number of peers selected by clients (related to tau) would be good.



[1] Briggs, C., Fan, Z., & Andras, P. (2020, July). Federated learning with hierarchical clustering of local updates to improve training on non-IID data. In 2020 international joint conference on neural networks (IJCNN) (pp. 1-9). IEEE.

[2] Caldas, S., Duddu, S. M. K., Wu, P., Li, T., Konečný, J., McMahan, H. B., ... & Talwalkar, A. (2019). Leaf: A benchmark for federated settings. In Workshop on Federated Learning for Data Privacy and Confidentiality

[3] Vahidian, S., Morafah, M., Chen, C., Shah, M., & Lin, B. (2023). Rethinking data heterogeneity in federated learning: Introducing a new notion and standard benchmarks. IEEE Transactions on Artificial Intelligence, 5(3), 1386-1397.

**Strengths And Weaknesses:**

Strengths:
 - The scope of the paper is clear
 - The experiments include several distribution shifts
 - A conclusion is drawn for each metric.

Weaknesses:
 - The difference between metrics is marginal, yet the deviation of the experiments is not highlighted.
 - All datasets are derived from traditional ML datasets.
 - Lack of building on related research

---

> ### Author Response · Authors · 2024-08-08
> **Answer to reviewer GnUc**
>
> Thank you for the insightful feedback on our paper. We address each of your points below:
>
> Regarding the weaknesses:
>
> "The difference between metrics is marginal, yet the deviation of the experiments is not highlighted."
>
> The suitability of different metrics depends on the specific problem at hand. We believe there are two interesting findings here:
>
> 1. Using any of these similarity metrics can improve the performance of FedAvg with random communication.
> 2. Identifying the best metric a priori is challenging, requiring careful investigation and tuning similar to other hyperparameters to determine the optimal choice for the problem.
>
> Additionally, the standard deviations for each experiment are indeed highlighted in Figures 3a, 4a, 4b, and 5abcd.
>
> "All datasets are derived from traditional ML datasets."
>
> We question why using traditional ML datasets is considered a weakness. Traditional ML datasets provide a standard benchmark that facilitates comparison with previous work.
>
> "Lack of building on related research."
> Could you specify which related research is lacking? Our work directly builds upon the DAC framework as proposed by [1] and studies a similar setup as in [2].
>
> [1] Edvin Listo Zec, Ebba Ekblom, Martin Willbo, Olof Mogren, and Sarunas Girdzijauskas. Decentralized
> adaptive clustering of deep nets is beneficial for client collaboration. In International Workshop on Trustworthy Federated Learning, pp. 59–71. Springer, 2022
>
> [2] Zexi Li, Jiaxun Lu, Shuang Luo, Didi Zhu, Yunfeng Shao, Yinchuan Li, Zhimeng Zhang, Yongheng Wang,
> and Chao Wu. Towards effective clustered federated learning: A peer-to-peer framework with adaptive
> neighbor matching. IEEE Transactions on Big Data, 2022.
>
> Requested changes:
>
> 1. Optimal Transport and Wasserstein Distance: We agree that optimal transport and Wasserstein distance are interesting topics. We can include a discussion on these metrics. However, one should note that estimating the probability distributions of each client poses a privacy challenge.
>
> 2. Hierarchical Clustering: The work by Briggs et al. (2020) involves hierarchical clustering in federated learning with a central server, requiring a global joint model for clustering. Our approach differs significantly as we operate in a fully decentralized system without a central server. Each client performs its own sampling and cluster assessment. Implementing hierarchical clustering in a decentralized system would likely necessitate extensive communication among all clients, leading to high communication costs.
>
> 3. If necessary, we can rerun our experiments using a dataset from Caldas et al. (2019) to include another dataset.
>
> 4. We do not use a random label distribution split in CIFAR-100. Clients are split into three distinct clusters based on the superclasses of the dataset, as described on page 7 of the paper.
>
> 5. The standard deviations for all experiments are shown in Figures 3a, 4a, 4b, and 5abcd. If needed, we can also add these to Table 2.
>
> 6. We will correct the ordering of Table 2 to match the subsections of the Results section.
>
> 7. We agree with the suggestion to discuss the number of peers selected by clients. We can rerun experiments varying $m$ and analyze the performance changes. We initially chose a small value to reflect realistic settings where clients can communicate with only a few peers each round (5% of total clients). Sampling too many clients might simplify the task unrealistically (and give rise to a high communication cost), which is why we avoided sampling with a large $m$.

---

### Review · Reviewer_nEmo · 2024-08-07

**Summary Of Contributions:**

The paper focuses on a decentralized adaptive clustering framework where $K$ clients train their local model for $E$ epochs and then each client merges its model with $m$ chosen peers through federated averaging. A random peer selection strategy works best when the data distribution across clients is IID. In the case of non-IID data, the convergence and performance of the model heavily depend on the peer selection. The main goal of this paper is to evaluate four different similarity metrics used for peer selection under a decentralized adaptive clustering framework. In particular, the paper considers the following four similarity metrics: 1. empirical loss, 2. cosine similarity on gradients, 3. cosine similarity on model weights, and 4. $L_2$ distance on model weights. The paper then presents FedSim which merges client models using a weighted average based on their similarities. The paper considers 4 different non-IID settings for the evaluation namely covariance shift, label shift, concept shift, and domain shift.

**Audience:**

Yes

**Claims And Evidence:**

No

**Requested Changes:**

1. Please add a pseudo code for the chosen setup. It will help the readers to better understand the decentralized adaptive clustering framework.
2. Do the clients need to compute the similarity at every communication round? Or can this be reduced to every $h$ round? It would be interesting to evaluate how the performance changes are we reduce the frequency of similarity evaluation (cluster reassignment ).
3. The paper deals with non-IID data but the experiments on conducted on the FedAvg algorithm which is meant for IID data. If the goal of the paper is to understand the impact of similarity metrics in non-IID settings then the analysis needs to extend beyond FedAvg. Consider decentralized learning algorithms meant for non-IID data such as QGM [1], NGM [2], etc.
4. Please improve the related works section by adding the literature on Federated Clustering Algorithms and Decentralized Learning Algorithms for heterogeneous data. Additionally, [3] might be relevant as it focuses on Data-heterogeneity-aware mixing.
5. It is useful to note the overheads of each metric. For example, inverse empirical loss evaluation requires additional $K$ forward passes through the model on the entire training dataset which is not feasible in a resource-constrained setting.

References:
1. Lin, Tao, et al. "Quasi-global momentum: Accelerating decentralized deep learning on heterogeneous data." arXiv preprint arXiv:2102.04761 (2021).
2. Aketi, Sai Aparna, Sangamesh Kodge, and Kaushik Roy. "Neighborhood Gradient Mean: An efficient decentralized learning method for non-iid data." Transactions on Machine Learning Research (2023).
3. Dandi, Yatin, et al. "Data-heterogeneity-aware mixing for decentralized learning." arXiv preprint arXiv:2204.06477 (2022).

**Strengths And Weaknesses:**

Strengths:
1. The paper evaluates the impact of 4 different similarity metrics under various non-IID distributions for decentralized adaptive clustering.
2. Experiments are conducted on multiple small-scale datasets.
3. The paper introduces a weighted averaging based on the clients' similarities referred to as FedSim.
4. The paper is clearly written and is easy to follow.

Weaknesses/questions:
1. This line of work is similar to Federated Clustering Algorithms [1-4] but the related work or the baselines doesn't include any Federated Clustering Algorithms.
2. Table 2 shows that FedSim doesn't have a significant benefit over FedAvg.
3. For the domain shift, why did you consider two completely different tasks? The typical domain shift scenarios consider the same tasks such as MNIST and SVHN.
4. For the label shift case where each cluster has a completely different set of classes, merging clients within the same cluster implies that the clients never learn about the classes present in other clusters. Is this intended behavior? In this case, the confusion matrix will be a better metric to report than accuracy.
5. Even though averaging only requires each client to communicate with $m$ peers, the computation of similarity requires each client to communicate with all the clients increasing communication and compute overhead.
6. For clarity: At the end of the training, there are $n$ unique models where $n$ is the number of clusters. Is the test accuracy averaged across all these models?
7. For FedSim, if the similarity of a given client is computed with respect to all the clients then why is the model averaging restricted only to $m$ peers?  How does the performance change if we do a similarity-based weighted averaging by merging all the clients i.e., $\sum_{k=1}^{K} s_k w_k^t$?

References:
1. Vardhan, Harsh, Avishek Ghosh, and Arya Mazumdar. "An Improved Federated Clustering Algorithm with Model-based Clustering." Transactions on Machine Learning Research (2024).
2. Avishek Ghosh, Jichan Chung, Dong Yin, and Kannan Ramchandran. An efficient framework for clustered federated learning. IEEE Transactions on Information Theory, 68(12):8076–8091, 2022. shorter version in NeurIPS 2021.
3. Yichen Ruan and Carlee Joe-Wong. Fedsoft: Soft clustered federated learning with proximal local updating. CoRR, abs/2112.06053, 2021.
4. Duan, Moming, et al. "Fedgroup: Efficient federated learning via decomposed similarity-based clustering." 2021 IEEE Intl Conf on Parallel & Distributed Processing with Applications, Big Data & Cloud Computing, Sustainable Computing & Communications, Social Computing & Networking (ISPA/BDCloud/SocialCom/SustainCom). IEEE, 2021.

---

> ### Author Response · Authors · 2024-08-07
> **Answer to reviewer nEmo**
>
> Thank you for the insightful feedback on our paper. We address each of your points below:
>
>
>
> 1. Our work differs fundamentally from the Federated Clustering Algorithms [1-4] in that we do not rely on a central server or a global model. Instead, each client maintains its own unique model, resulting in $K$ unique models across the system at all timesteps. This decentralization sets our approach apart.
>
>
> 2. We acknowledge that FedSim does not consistently outperform FedAvg across all tasks. We consider this an interesting finding that highlights the robustness of FedAvg, particularly in scenarios involving distributional shifts and non-iid data. FedSim is beneficial when communication is random because it reduces the impact of "wrong" clients. This is especially true in the concept shift case, where the conditional distributions differ between clusters. However, when using similarity metrics for sampling, the advantage of FedSim diminishes, making FedAvg a sufficient approach. This is a key insight from our study.
>
> 3. We intentionally chose two completely different tasks to study domain shift because we wanted to explore scenarios where merging with the wrong cluster incurs a significant negative cost for each client. In typical examples like MNIST/SVHN, even if a client merges with the wrong cluster, the impact is minimal due to the relative similarity of the distributions. Our approach allows us to evaluate how well similarity metrics prevent incorrect merges.
>
> 4. Indeed, in our approach the goal is for each client to learn only the labels present in its own cluster. At test time, clients are evaluated on a test set containing only their specific cluster labels. We believe that adding a confusion matrix may not significantly enhance the information, but we can include it if deemed necessary for clarity.
>
> 5. We apologize if this was unclear. Clients do not compute similarities for all $K$ clients. Instead, they maintain similarity scores only for clients they have communicated with up to a given timestep $t$. Following the approach by Listo Zec et al. [5], clients approximate similarities based on past communications, which prevents any increase in communication or computational overhead.
>
> 6. There seems to be a misunderstanding here. Unlike typical federated learning setups with a central server, we have $K$ models, one for each client (not each cluster). Test accuracies are averaged over all clients, with each client evaluated on its specific test set according to its cluster.
>
> 7. We limit model averaging to $m$ peers because sampling $m=K$ peers in every communication round is infeasible. Clients only have similarity metrics for the $m$ peers they sample, not for all clients.
>
> If you have further questions or require additional clarifications, please let us know. We appreciate your valuable feedback and the opportunity to improve our work.
>
> References:
>
> [5] Listo Zec, et al. "Decentralized adaptive clustering of deep nets is beneficial for client collaboration." International Workshop on Trustworthy Federated Learning. Cham: Springer International Publishing, 2022.
>
>
> Requested changes:
>
> 1. We will include pseudocode for the algorithm to enhance clarity and understanding.
>
> 2. At each timestep, we evaluate the similarity of the $m$ sampled peers because we have access to them. The most computationally intensive metric is the inverse loss, which requires a forward pass for each model. The other metrics are computationally inexpensive as they only involve comparing distances between model parameters.
>
> 3. Our work is distinct from QGM and NGM, which focus on consensus optimization. Instead, we address the issue of varying data distributions across clusters. In QGM and NGM, it is implicitly assumed that a single model can achieve optimal risk across all clients, a common assumption also in federated learning. In contrast, we do not make this assumption, as each client in our study maintains a unique model. Cluster distributions can be orthogonal, where a single model cannot fit both distributions (e.g. different conditional distributions).
>
> 4. We will update the related work section to include the relevant literature you suggested.
>
> 5. We will detail the communication costs associated with each metric, emphasizing that the empirical loss metric is more costly than the others.

---

> ### Comment · Reviewer_nEmo · 2024-08-07
> **Reply from Reviewer nEmo**
>
> Thanks for the clarification. It is now clear to me that each client samples a different set of $m$ peers at each iteration based on the similarity vector $s_i$  and therefore by the end of the training, there are potentially $K$ unique models. The similarity values at client $i$ corresponding to the $m$ chosen peers are updated at that communication round.
>
> Follow up:
> 1. I agree that Federated Clustering Algorithms are designed for setups with a central server. However, the metrics used to form the cluster can be trivially extended to the DAC framework. For example, [1] uses $L_2$ distance. Therefore, this is a relevant line of work to be considered for the related work, and is important to point out the issues that arise with extending Federated Clustering Algorithms to DAC setups.
> 2. "FedSim is beneficial when communication is random" - FedSim results with random sampling are presented in Table. 2. Can you please point out the relevant results for this claim?
> 3. How is the $s_i$ vector initialized on each client? For a given client $i$, does $s_i$ converge to a stationary distribution?
> 4. What is the value of $E$ (local steps) used for the experiments?

---

> ### Author Response · Authors · 2024-08-08
>
> Thank you for your follow-up questions and the fast reply!
>
> 1. We already cite [1] in our paper. We agree that it is important to also cite and discuss the follow-up work [2], which is relevant for understanding how the metrics used in Federated Clustering Algorithms can be extended to the DAC framework.
>
> [1] Avishek Ghosh, Jichan Chung, Dong Yin, and Kannan Ramchandran. An efficient framework for clustered federated learning. IEEE Transactions on Information Theory, 68(12):8076–8091, 2022. shorter version in NeurIPS 2021.
>
> [2] Vardhan, Harsh, Avishek Ghosh, and Arya Mazumdar. "An Improved Federated Clustering Algorithm with Model-based Clustering." Transactions on Machine Learning Research (2024).
>
> 2. Our previous claim was indeed unclear. We do not perform random sampling with FedSim. The "Random" baseline refers to random communication combined with FedAvg for aggregation. FedSim shows significant benefits when aggregating with incorrectly sampled clients can be highly detrimental, such as when the data distributions between clusters are vastly different. In these scenarios, FedSim can down-weight "wrong" clients, unlike FedAvg. This is particularly evident in the concept shift task, shown in Figure 3a and Table 2, where FedSim outperforms FedAvg by a large margin. For the domain shift experiment using two different datasets (FashionMNIST and MNIST), FedAvg with similarity sampling performs quite well. A key takeaway is that FedAvg, with a proper similarity metric for sampling and appropriate hyperparameter tuning, can effectively handle distributional shifts and non-iid data. This is demonstrated in Table 2, where FedAvg consistently outperforms the Random baseline and often surpasses FedSim.
>
> 3. The $s_i$ vectors are initialized with a uniform prior over all clients. After the first sampling, these vectors are updated based on the newly sampled information. Different clients converge to different $s_i$ values, as partially shown in the heatmaps in Figure 6 of the appendix. If needed, we can also provide the heatmap of the final $s_i$ values at time $T$, which closely resembles the heatmap.
>
> 4. We performed hyperparameter tuning using grid search, with $E$ being one of the parameters. The optimal value was found to be $E=1$ based on validation accuracy. While longer local training reduces communication costs, it also decreases performance. We report all relevant hyperparameters in Table 8 of the appendix. Additionally, we use the Adam optimizer locally on all clients.

---

### Review · Reviewer_e9iP · 2024-11-14

**Summary Of Contributions:**

The paper titled "On the Effects of Similarity Metrics in Decentralized Deep Learning Under Distributional Shift" studies decentralized learning (DL) under heterogeneous client data distributions, aiming to enhance client collaboration without a central server and direct data sharing. Here’s a mathematical summary:

   Each client \(i\) seeks to minimize an empirical risk over its dataset \(\mathcal{D}_i\), represented as:
   \[
   w_i^* = \arg \min_{w_i \in \mathbb{R}^d} \mathbb{E}_{(x, y) \sim \mathcal{D}_i} \left[ \ell(f_i(w_i; x), y) \right]
   \]
   where \(\ell\) is a local loss function, \(f_i\) the model, and \(w_i\) the parameters.

   Clients in a fully decentralized network use similarity metrics to identify peers. Probability vector \(p_i\) for selecting a peer \(j\) is defined via:
   \[
   p_i(s_i) = \frac{e^{\tau s_i}}{\sum_{k=1}^K e^{\tau s_{ik}}}
   \]
   where \(s_{ik}\) is the similarity between clients \(i\) and \(k\), and \(\tau\) controls selection sensitivity.

   The paper evaluates four metrics, including:
   - nverse Empirical Loss: \(s_{ik} = \left( \sum_{(x, y) \in \mathcal{D}_k} \ell(w_i; x, y) \right)^{-1}\)
   - Inverse \(L_2\) Distance: \(s_{ik} = \|w_i - w_j\|_2^{-1}\)
   - Cosine Similarity of Weights: \(s_{ik} = \frac{w_i \cdot w_j}{\|w_i\| \|w_j\|}\)
   - Cosine Similarity of Gradients: \(g_i^t = w_i^t - w_i^0\) and \(s_{ik} = \frac{g_i \cdot g_j}{\|g_i\| \|g_j\|}\)

   A proposed FedSim adjusts the FedAvg aggregation by weighing models by similarity:
   \[
   w^{t+1} = \sum_{k=1}^m s_k w_k^t
   \]
   where \(s_k\) is the similarity of client \(k\), normalized to sum to one.
   Testing these formulations on benchmark datasets (CIFAR-10, Fashion-MNIST, etc.), the study highlights that cosine similarity metrics consistently offer robust performance across scenarios with distributional shifts.

This formulation-based analysis highlights how different similarity metrics impact decentralized learning under non-IID conditions and supports the effectiveness of FedSim under complex, heterogeneous data environments.

**Audience:**

Yes

**Broader Impact Concerns:**

N.A.

**Claims And Evidence:**

Yes

**Requested Changes:**

See above discussion.

**Strengths And Weaknesses:**

1.
   The paper's objective of the formulation does not account for the fact that **heterogeneous data distributions** can make this local objective conflict with other clients’ objectives, particularly when \(P_i(x, y) \neq P_j(x, y)\). Without explicit constraints or regularization terms in the formulation to align clients' objectives, **theoretical convergence of decentralized learning is not guaranteed**. A regularization term, as used in federated learning (e.g., penalizing the distance between local parameters \(w_i\) and a global parameter \(w\)), could help improve alignment. Thus, the soundness of assuming this formulation is suitable for decentralized non-IID settings may be questioned.

2.
   One similarity metric proposed in the paper is the inverse empirical loss:
   \[
   s_{ik} = \left( \sum_{(x, y) \in \mathcal{D}_k} \ell(w_i; x, y) \right)^{-1}
   \]
   There are concerns with this metric’s soundness:
   - Noise Sensitivity: If \(\ell(w_i; x, y)\) is small, \(s_{ik}\) could be excessively large, leading to **unstable client selection** due to the inverse relationship. Outliers in empirical loss values can disproportionately influence peer selection, causing clients to potentially choose suboptimal collaborators.
   - Dependency on Data Quality and Size: \(s_{ik}\) depends on the dataset \(\mathcal{D}_k\) used for evaluation. If \(|\mathcal{D}_k|\) is small or not representative, then \(s_{ik}\) may not reliably reflect the true similarity between clients \(i\) and \(k\), leading to inaccurate cluster formation.

   For robustness, a **softmax scaling** or **bounded similarity metric** might improve stability, especially in low-data contexts.

3.
   The FedSim approach modifies the traditional FedAvg by weighting client updates based on similarity:
   \[
   w^{t+1} = \sum_{k=1}^m s_k w_k^t
   \]
   Here, \(s_k\) represents a similarity weight. However:
   -  The paper assumes normalization of \(s_k\) but does not analyze whether the weighted averaging preserves convergence properties (e.g., as in FedAvg with uniform weights). Without this analysis, there’s no guarantee that FedSim will converge to an optimal or even stable solution in non-IID settings.
   - Impact of Misclassification in Similarity: FedSim may amplify errors if the similarity metrics misclassify clients, assigning high similarity to incompatible models, thereby affecting model quality. **Error propagation** in FedSim could be more severe than in FedAvg due to similarity-weighted averaging.

4.
   For selecting peers, the paper employs a softmax function with an inverse temperature parameter \(\tau\):
   \[
   p_i(s_i) = \frac{e^{\tau s_i}}{\sum_{k=1}^K e^{\tau s_{ik}}}
   \]
   This choice raises several concerns:
   - The effectiveness of similarity-based clustering is highly sensitive to \(\tau\). For large \(\tau\), the softmax approximates an \(\arg\max\) function, potentially causing instability if similarity estimates are noisy. For small \(\tau\), similarity differences are minimized, reducing clustering effectiveness.
   - The paper does not specify a mechanism to adapt \(\tau\) across rounds, which could lead to **suboptimal peer selection** if data heterogeneity or the number of available clients varies over time. A theoretical study of optimal \(\tau\) or an adaptive approach would improve the robustness.

---

> ### Author Response · Authors · 2024-11-14
>
> We thank the reviewer for the insightful feedback.
>
> 1. We agree that diverse data-generating processes across clients may indeed result in distinct objectives (note also that there is no global model in our setup, thus regularization towards such a solution is not possible). Our study directly addresses this by empirically analyzing the impact of various similarity metrics under such conditions, as seen across our experiments. The synthetic experiment, for example, demonstrates a scenario with concept shift, where clusters exhibit differences in $p(y|x)$, providing a clear instance of conflicting objectives. The primary aim of our work is to explore how the choice of similarity metric influences learning in these decentralized settings. As shown in Figure 3a, cosine similarity (computed on gradients or weights) is robust, while L2 and inverse loss metrics are less effective because they frequently draw models from unrelated clusters, thereby hindering learning. In cases of label and covariate shifts, the choice of similarity metric remains important but has a less pronounced impact. This is due to the consistent $p(y|x)$ across clusters, making occasional sharing outside of the "correct" cluster less detrimental.
>
> 2. We agree with the concerns you bring regarding empirical loss. As we discuss in the paper, this metric performs best when datasets are large and of high quality, as it can otherwise become a noisy measure of similarity. This is evident in our experiments, particularly in Figure 3b, where we show that inverse empirical loss and L2 metrics are indeed more sensitive to sample size fluctuations, underperforming compared to the more robust cosine similarity metric. These observations reinforce the notion that the inverse empirical loss metric may not be ideal in low-data contexts and highlights the need for alternative metrics or bounded scaling techniques for settings where data variability could distort client selection. However, developing new similarity metrics is beyond the scope of this paper; our focus is on investigating commonly used metrics in decentralized learning.
>
> 3. We acknowledge that the theoretical analysis of convergence properties under similarity-weighted averaging is an important area for further study. As our paper primarily focuses on the empirical effects of various similarity metrics, we prioritized experimental analysis over theoretical convergence proofs. This decision was made to first understand the practical implications of similarity-weighted updates in decentralized learning, where challenges like non-iid data and client misalignment are pervasive. Nonetheless, our empirical results across multiple settings, including concept, label, covariate, and domain shifts, show that FedSim can outperform traditional FedAvg in scenarios where inappropriate merges could degrade learning. However, an interesting finding which we discuss is that FedAvg is relatively robust in covariate shift and label shift settings. Regarding FedSim, we acknowledge that misclassification in similarity could amplify errors in the algorithm. However, this is exactly why we underscore the importance of selecting an appropriate similarity metric to mitigate this risk. We agree that interesting future works include a theoretical analysis to further support these empirical findings, focusing on bounding the error introduced by misclassified similarities and exploring regularization techniques to enhance FedSim’s resilience to such cases.
>
> 4. We recognize that $\tau$ plays a critical role in balancing the selectivity of peer sampling: higher values of $\tau$ emphasize the strongest similarities by approximating an argmax function, while lower values smooth the selection process by minimizing similarity distinctions. To address this sensitivity, we treat $\tau$ as a key hyperparameter and tune it empirically through validation, similar to other hyperparameters such as learning rate and batch size. In our study, we experimented with a range of values and selected the one that yielded the best validation performance for each specific scenario (see Appendix). Our empirical results, shown in the paper, illustrate that careful tuning of helps achieve stable clustering and effective peer selection, even under varying levels of data heterogeneity and noise. $\tau$ can absolutely be adapted during rounds, for example by increasing it as rounds pass - leading to more exploitation in the end of training and more exploration in the beginning. This was studied by [1].
>
> Thank you again for these valuable insights, which contribute to the further refinement of our approach.
>
> [1] Zec, E. L., Ekblom, E., Willbo, M., Mogren, O., & Girdzijauskas, S. (2022, July). Decentralized adaptive clustering of deep nets is beneficial for client collaboration. In International Workshop on Trustworthy Federated Learning (pp. 59-71). Cham: Springer International Publishing.

---

> ### Author Response · Authors · 2024-11-14
>
> Expanding upon your point 3) regarding theoretically justifying FedSim:
>
> Under a perfect similarity metric, $s_{ij}$ approaches 1 when $P_i \approx P_j$​ and 0 otherwise. Therefore, the aggregated model converges towards a model trained predominantly on data from distributions similar to $P_j$. This weighted aggregation thus approximates a local optimal solution for client $j$, minimizing the expected loss over similar data distributions.
>
>
> $ E_{(x, y) \sim P_j} \left[ \ell\left( \sum_{i=1}^m s_{ij} w_i; x, y \right) \right] \approx \sum_{i=1}^m s_{ij} E_{(x, y) \sim P_j} \left[ \ell\left( w_i; x, y \right) \right].$
>
> Although developing such an optimal similarity metric theoretically is beyond the scope of this paper, we believe as you point out that this is a promising direction for future research, possibly drawing on theories from optimal transport. For now, our empirical studies investigate the effectiveness of commonly used similarity metrics, highlighting their strengths and limitations.
>
> We will update our paper discussing this.

---

> > ### Comment · Reviewer_e9iP · 2024-12-17
> >
> > Thank you to the authors for addressing my concern, which has clarified the paper.

---

### Author Response · Authors · 2024-11-22
**Response to reviewers**

We sincerely thank all reviewers for their time and effort in evaluating our work and for their valuable feedback, which has helped us enhance our work. We have carefully addressed each reviewer's comments and believe that our responses effectively address the concerns raised. For your convenience, all changes in the revised manuscript are highlighted in blue text.

Summarization of key changes:

- Included pseudocode for the algorithm to enhance clarity and understanding.

- We have detailed the communication costs associated with each metric, emphasizing that the empirical loss metric is more costly than the others.

- We have added a discussion in the future work section on optimal transport and the Wasserstein distance.

- We have corrected the ordering of Table 2 to match the subsections of the Results section.

- We have added standard deviations to Table 2

- We have run extra experiments on varying number of sampled clients $m$ for the CIFAR-10 two-cluster setup. See results of these experiments in Figure 6 in the Appendix, where we also discuss it.

- We have added a justification of FedSim's aggregation method in Section 3.5.

We have also added the following to the related work or discussion:

 - An Improved Federated Clustering Algorithm with Model-based Clustering, Vardhan et al.
 - Fedgroup: Efficient federated learning via decomposed similarity-based clustering, Duan et al.
 - Fedsoft: Soft clustered federated learning with proximal local updating, Ruan et al.
 - Quasi-global Momentum: Accelerating Decentralized Deep Learning on Heterogeneous Data, Lin et al.
 - Neighborhood gradient mean: An efficient decentralized learning method for non-iid data, Aketi et al.
 - Data-heterogeneity-aware mixing for decentralized learning, Dandi et al.
 - Computational optimal transport: With applications to data science, Peyré et al.
 - Federated learning with hierarchical clustering of local updates to improve training on non-IID data, Briggs et al.

---

### Comment · Reviewer_GnUc · 2024-11-25
**Limitations of empirical study**

Dear Authors,

First of all, I would like to apologize for my late reply, I didn't see the notifications until the 3rd review arrived last week.

After considering all reviews and author responses my main concern is still this:
The empirical evaluation of the proposed work consists of comparing to only the most straightforward decentralized method (FedAvg) and the proposed method's advantage over it is marginal.

Both Reviewer nEmo and GnUc (me) addressed this concern by suggesting relevant, but federated papers that can be adapted to decentralized setting. The authors argued against using these, but didn't provide other methods to compare to.

I see the contributions of the proposed work twofold:

1) Comparing the effect of choosing different similarity metrics. In this contribution, I do not see why the proposed methods should be excluded from the experiments, given that choosing similarity metrics is relevant to a federated setting as well as to a decentralized setting.

2) The proposed FedSim as an alternative method to existing ones. In this case, I do not think comparing to only FedAvg is enough to put the method into context. If there is closely related work, the authors should make effort implement them and if there isn't, the authors should consider adapting the next best thing.

---

> ### Comment · Reviewer_GnUc · 2024-11-25
>
> Another arguably related Federated Learning papers where the server selects a subset of client according to a rule, and can be applied in distributed learning with the same rule at every client:
>
> Yae Jee Cho, Jianyu Wang, and Gauri Joshi. Towards under-
> standing biased client selection in federated learning. In In-
> ternational Conference on Artificial Intelligence and Statis-
> tics, pages 10351–10375. PMLR, 2022.
>
> Jianyu Wang, Qinghua Liu, Hao Liang, Gauri Joshi, and
> H Vincent Poor. Tackling the objective inconsistency prob-
> lem in heterogeneous federated optimization. Advances
> in neural information processing systems, 33:7611–7623,
> 2020
>
> Wu, Hongda, and Ping Wang. "Node selection toward faster convergence for federated learning on non-iid data." IEEE Transactions on Network Science and Engineering 9.5 (2022): 3099-3111.

---

> ### Author Response · Authors · 2024-11-25
>
> Dear Reviewer,
>
> Thank you for your thoughtful and detailed feedback. We appreciate your engagement and the opportunity to clarify and strengthen our work.
>
> We acknowledge your concern regarding the comparison to only FedAvg with gossip learning (random communication). Our primary aim was to explore the potential of similarity-based peer selection within the decentralized learning framework, which, to our knowledge, is an area with limited existing methods beyond standard gossip algorithms using FedAvg. This makes direct comparisons challenging.
>
> In our work, we focused on how similarity metrics can be used within the constraints of decentralized learning **without a central server**. Including federated methods that rely on central coordination would not provide a fair comparison, as they operate under different assumptions.
>
> We appreciate your observation that FedSim can be implemented in a centralized federated setting and compared to the mentioned literature. Our current work, however, is specifically focused on investigating how similarity metrics affect client identification within a decentralized learning framework. Including comparisons with centralized FL methods would shift the focus away from the decentralized aspects we aim to explore. Moreover, the mechanisms and challenges in decentralized settings differ significantly from those in centralized ones, making direct comparisons less straightforward. That said, we agree that extending our analysis to centralized FL is an intriguing research direction. We are excited about the potential insights this could bring and plan to explore this in our future work.
>
> Regarding the FL papers you mention:
>
> - Wang et al. (FedNova, 2020): FedNova addresses objective inconsistency using normalization steps performed by a central server. This central coordination is crucial for their algorithm and is not feasible in a decentralized setup without significant alterations that would fundamentally change the algorithm. We appreciate this reference and see potential in exploring decentralized normalization techniques in future work.
>
> - Cho et al. (2022): Their method involves a central server that biases client selection towards those with higher local losses. The goal is to learn a single global model $w$ for all clients and to efficiently sample clients to optimize this global objective. This approach differs fundamentally from our goal, which is to solve $K$ different optimization problems and to identify the most beneficial collaborators for each individual client. In our setup, sampling clients with the highest loss does not make sense because we are not aiming for a shared global model.
>
> - Wu et al. (2022): This approach involves a central server selecting nodes based on the relationship between local and global gradients. Without access to global gradients or a central entity, implementing this method in a decentralized environment is not straightforward.
>
> Adapting these methods to a decentralized context would require significant modifications and could result in entirely new algorithms, which is beyond the scope of our current work. Our aim was to explore the potential of similarity-based peer selection within the existing decentralized framework and demonstrate its effectiveness over standard gossip-based methods.

---

> > ### Comment · Reviewer_GnUc · 2024-11-25
> >
> > Dear Authors,
> >
> > I am afraid I do not understand the required significant modifications.
> >
> > If a federated learning algorithm requires the central server to run $\mathcal{A}$ algorithm on weights $w_j, j\in\mathcal{M}=\{c_1,c_2,...,c_m\}$ to attain global model $\overline{w}$ and global selection probabilities $\mathbf{p}$ such that $(\overline{w},\mathbf{p})\leftarrow\mathcal{A}(\{w_1,...,w_j\}),j\in\mathcal{M}$ , certainly a given client $c_i$ that can communicate with any client in a fully connected setting such as in your experiment setup, this client can use the same $\mathcal{A}$ algorithm such that $(w_i,\mathbf{p_i})\leftarrow\mathcal{A}(\{w_1,...,w_j\}),j\in\mathcal{M}_i$.
> >
> > In fact, a client in your decentralized setting can do everything a central server in a federated setting can do, and even more, because a federated aggregation does not require a server-side dataset, while a client-as-server has a $\mathcal{D}_i$ local dataset.
> >
> > If we apply a cosine similarity metric in a fully connected $K_{m+1}$ graph, each client is required to send their $w_i$ weights to every $j\neq i$ client to similarity check and vice versa. Therefore, in this toy example, the distributed learning setting is communicating exactly the same as all $m+1$ clients acting as central servers in their own federated setting.

---

> > > ### Author Response · Authors · 2024-11-25
> > >
> > > Thank you for your comment. We would like to take this opportunity to address the misunderstanding regarding decentralized learning and its differences from federated learning, even in a fully connected graph.
> > >
> > > While it is theoretically possible for clients in a fully connected decentralized network to communicate with all other clients, effectively performing operations similar to a central server in federated learning, there are practical limitations that make this approach challenging in decentralized settings:
> > >
> > > 1. **Centralized coordination vs. decentralized learning.**
> > > In FL, the central server aggregates model updates and computes metrics (e.g., cosine similarity, selection probabilities). The central server has a global view of all participating clients’ states and can coordinate learning with this information. In contrast, in decentralized learning, each client operates autonomously and only has access to information from its neighbors in the communication graph. This lack of global state fundamentally changes the dynamics of both learning and communication. In a fully connected graph, while every client may theoretically communicate with others, this requires peer-to-peer interactions for every pair of clients, leading to **quadratic communication overhead**  with respect to the number of clients $K$ when scaling the network. For example, with 1,000 clients, there would be nearly a million communication exchanges per round. This level of communication is not practical in real-world scenarios where network bandwidth, latency, and energy consumption are critical factors. Decentralized learning aims to minimize communication costs by limiting interactions to a subset of peers. This approach ensures scalability and makes decentralized systems viable for large-scale deployments. FL leverages a central server to reduce this to $\mathcal{O}(K)$ communication complexity, as clients only communicate with the server.
> > >
> > > 2. **Resource constraints.** Clients in decentralized networks often have limited computational and storage resources, especially in edge computing scenarios (e.g., smartphones, IoT devices). Expecting each client to perform the role of a central server places a heavy burden on these devices.
> > >
> > > 3. **Objective Differences**. In our work, the goal is to solve $K$ different optimization problems, focusing on personalized models for each client by identifying beneficial collaborators. FL algorithms like those mentioned aim to learn a **single global model**, which is a fundamentally different objective.
> > >
> > > 4. **Redundancy**.  If every client acts as a central server and aggregates models from all other clients, this redundancy can be inefficient. However, decentralized learning algorithms (e.g. Gossip Learning) are typically designed to avoid such redundancy by structuring communication and aggregation more efficiently.
> > >
> > > While we acknowledge that, in theory, clients could perform centralized operations in a fully connected graph, the practical implications and fundamental differences in objectives make such an approach unsuitable. Adapting federated algorithms to decentralized systems without significant modifications would be challenging and could compromise the advantages of decentralization.
> > >
> > > We hope this clarifies our position and we are grateful for your feedback, which encourages a deeper discussion on the nuances between decentralized and federated learning.

---

> ### Author Response · Authors · 2024-11-25
> **Some added clarification to reviewer GnUc**
>
> We recognize that aggregation methods focusing on choosing the weights $\alpha_i$ in the model aggregation step $\alpha_i$ in $\bar{w} = \sum_i \alpha_i w_i$ can be implemented within our framework and compared with approaches like FedAvg or FedSim. Our decentralized setting indeed allows for different aggregation schemes, and we have proposed FedSim as an alternative in our paper.
>
> However, we want to emphasize that the primary contribution of our work is investigating the effectiveness of similarity metrics for client identification in a decentralized learning environment. Our main focus is on how different similarity metrics impact peer selection and, consequently, the performance of decentralized learning algorithms. While aggregation methods are an important aspect, they are a secondary contribution in our study.
>
> To our knowledge, there are no aggregation algorithms that have been proposed in FL that attempt to mitigate the bad affects from aggregating with clients from "wrong" clusters. If there are, we are definitely able to rerun experiments with these aggregation schemes (if they do not require any global information), although it will take some time for the experiments to finish.

---

### Decision · Action_Editor_wJ8T · 2025-01-22

**Recommendation:** Accept with minor revision

**Comment:**

The reviewers’ opinions on this paper are divided.

One reviewer argued for acceptance, while two reviewers raised concerns about the limited scope of the empirical evaluation. Specifically, they noted that the synthetic scenarios (e.g., distribution shifts and network topology) address only a narrow range of cases. They also emphasized that, given the established nature of personalized learning, additional benchmarks would have been necessary to better assess the potential impact of this work.

Despite these concerns, I believe that the submission meets TMLR's acceptance criteria. The paper offers a focused contribution to the field of personalized learning, and I therefore recommend acceptance.


---
Minor revision:
- In the newly added Section 3.1, the computational cost of the metrics is discussed. However, it is unclear why stochastic quantities are being compared to deterministic measures. For example, cosine similarity could also be computed using full gradients, while the empirical loss could be stochastically estimated. Please clarify this discussion for the final version.

**Audience:**

The claims investigated in this paper are confined to the limited scope outlined above.

Two reviewers expressed concerns that the findings may lack broader appeal due to the narrow scope of the experiments and the limited selection of baselines. However, one reviewer found the results to be interesting and relevant.

In my view, this is a very borderline case. Nevertheless, given that personalized learning is a subfield of interest within the TMLR community, I believe the results could still be valuable to certain members of the audience.

**Claims And Evidence:**

This paper investigates the impact of four similarity metrics for client sampling within the decentralized adaptive clustering (DAC) framework proposed by Listo Zec et al. (2022). The study evaluates these metrics across six experimental settings, simulating various synthetically generated distributional shifts.

Within this scope, the claims made in this paper are supported by numerical evidence.

Edvin Listo Zec, Ebba Ekblom, Martin Willbo, Olof Mogren, and Sarunas Girdzijauskas. Decentralized
adaptive clustering of deep nets is beneficial for client collaboration. In International Workshop on Trustworthy Federated Learning, pp. 59–71. Springer, 2022.